# Application of In Vivo Imaging Techniques and Diagnostic Tools in Oral Drug Delivery Research

**DOI:** 10.3390/pharmaceutics14040801

**Published:** 2022-04-06

**Authors:** Stefan Senekowitsch, Philipp Schick, Bertil Abrahamsson, Patrick Augustijns, Thomas Gießmann, Hans Lennernäs, Christophe Matthys, Luca Marciani, Xavier Pepin, Alan Perkins, Maximilian Feldmüller, Sarah Sulaiman, Werner Weitschies, Clive G. Wilson, Maura Corsetti, Mirko Koziolek

**Affiliations:** 1Department of Biopharmaceutics and Pharmaceutical Technology, Center of Drug Absorption and Transport, University of Greifswald, 17489 Greifswald, Germany; stefan.senekowitsch@uni-greifswald.de (S.S.); philipp.schick@uni-greifswald.de (P.S.); sager.max@googlemail.com (M.F.); werner.weitschies@uni-greifswald.de (W.W.); 2Oral Product Development, Pharmaceutical Technology & Development, Operations, AstraZeneca, 431 83 Gothenburg, Sweden; bertil.abrahamsson@astrazeneca.com; 3Department of Pharmaceutical and Pharmacological Sciences, Drug Delivery and Disposition, KU Leuven, 3000 Leuven, Belgium; patrick.augustijns@kuleuven.be; 4Boehringer Ingelheim Pharma GmbH & Co., KG, 88400 Biberach, Germany; thomas.giessmann@boehringer-ingelheim.com; 5Department of Pharmaceutical Biosciences, Translational Drug Discovery and Development, Uppsala University, 752 36 Uppsala, Sweden; hans.lennernas@farmbio.uu.se; 6Department of Chronic Diseases and Metabolism, KU Leuven, 3000 Leuven, Belgium; christophe.matthys@uzleuven.be; 7Department of Endocrinology, University Hospitals Leuven, 3000 Leuven, Belgium; 8NIHR Nottingham Biomedical Research Centre, Nottingham University Hospitals NHS Trust, Nottingham NG7 2UH, UK; luca.marciani@nottingham.ac.uk (L.M.); alan.perkins@nottingham.ac.uk (A.P.); sarah.sulaiman@nottingham.ac.uk (S.S.); maura.corsetti@nottingham.ac.uk (M.C.); 9Nottingham University Hospitals NHS Trust and Translational Medical Sciences, School of Medicine, University of Nottingham, Nottingham NG7 2UH, UK; 10New Modalities and Parenteral Development, Pharmaceutical Technology & Development, Operations, AstraZeneca, Macclesfield SK10 2NA, UK; xavier.pepin@astrazeneca.com; 11Radiological Sciences, School of Medicine, University of Nottingham, Nottingham NG7 2UH, UK; 12Bayer AG Pharmaceuticals, 13353 Berlin, Germany; 13Strathclyde Institute of Pharmacy and Biomedical Sciences, University of Strathclyde, Glasgow G4 0RE, UK; clivegwilson@mac.com; 14NCE Formulation Sciences, AbbVie Deutschland GmbH & Co. KG, 67061 Ludwigshafen, Germany

**Keywords:** biopharmaceutics, imaging tools, in vivo, pharmacokinetics, aspiration, salivary tracer technique, magnetic marker monitoring, high-resolution manometry, telemetric capsules, scintigraphy, magnetic resonance imaging

## Abstract

Drug absorption following oral administration is determined by complex and dynamic interactions between gastrointestinal (GI) physiology, the drug, and its formulation. Since many of these interactions are not fully understood, the COST action on “Understanding Gastrointestinal Absorption-related Processes (UNGAP)” was initiated in 2017, with the aim to improve the current comprehension of intestinal drug absorption and foster future developments in this field. In this regard, in vivo techniques used for the characterization of human GI physiology and the intraluminal behavior of orally administered dosage forms in the GI tract are fundamental to gaining deeper mechanistic understanding of the interplay between human GI physiology and drug product performance. In this review, the potential applications, advantages, and limitations of the most important in vivo techniques relevant to oral biopharmaceutics are presented from the perspectives of different research fields.

## 1. Introduction

The successful development of orally administered drugs requires a thorough understanding of human gastrointestinal (GI) physiology, together with the biopharmaceutical behavior of oral drug products in the human GI tract. Despite considerable progress in formulation development, increasing knowledge about the physiology and biochemistry of the GI tract, and more powerful in vitro and in silico oral biopharmaceutics tools, it is often difficult to make reliable assumptions about the in vivo behavior of new drug formulation approaches. Therefore, in vivo studies in humans still provide the only way to assess the performance of a novel drug product. Pharmacokinetic (PK) studies with healthy adults are still often used in this respect. The combination of PK studies with in vivo techniques—such as scintigraphy or magnetic marker monitoring (MMM)—has contributed to a deeper understanding of drug release and regional absorption mechanisms, and is particularly helpful at later stages of drug product development. Before these are described in more detail in the following sections, we will take a closer look at the challenges of PK studies in early drug product development.

Early clinical trials (phase I trials) evaluate the safety, tolerability, and pharmacokinetics of development compounds. Although safety aspects have the highest priority in a trial of healthy subjects, PK data support the examination of key processes relevant to early drug development:*1.* *Which dose provides the expected therapeutic concentration time profile?**2.* *Which dosing regimen is required in order to maintain this concentration?**3.* *Can these target concentrations be achieved with standard formulations?**4.* *Is the drug exposure dose relationship linear?**5.* *What recommendations can be given to the patients regarding food intake?*

To exert an effect, and depending on the mechanisms of action, a drug must reach an effective concentration in the target tissue, sustained for sufficient time in order to trigger a pharmacodynamic response. For most drug presentations, the point of administration is far from the site of action. The oral route remains the most convenient route of administration for drug products that can be absorbed via the GI tract. In contrast to the intravenous route of drug administration, the processes of drug release, dissolution, and absorption are crucial to determining the drug plasma concentration time profile and, thus, the onset and duration of the pharmacological action. Depending on the drug’s mechanism of action and biopharmaceutical properties, oral dosage forms must meet different requirements. For instance, if a constant plasma concentration level over a long period is targeted, the release of the drug often has to be slowed down by the utilization of modified release dosage forms. In the event of low solubility of a drug candidate, which is often associated with restricted bioavailability, the use of suitable formulation strategies may improve the performance of oral dosage forms.

The oral route allows the use of a wider range of acceptable pharmaceutical excipients, which can help to resolve issues of drug wettability or solubility, and/or control the drug release to modify the drug absorption rate, thereby providing a better control of the drug plasma concentration time profile, in order to minimize side effects, prolong efficacy, and reduce dosing frequency. Excipients and choices of polymorphs or salts for the drug moiety can also contribute to reducing the effects of food or acid-reducing agents on the drug’s pharmacokinetics, providing for more safety and convenience to the patient.

For the PK profile of an orally administered drug product, the prandial state can be of major importance. Food intake can alter the oral bioavailability of drugs, and these so-called food effects are often highly variable and poorly predictable. Therefore, they have to be investigated for each development compound. According to the Food and Drug administration (FDA) guidance on food-effect studies from 2002, the drug product should be given 30 min after start of the test meal, which is a high-fat and high-calorie meal of 800–1000 kcal with about 150, 250, and 500–600 kcal derived from protein, carbohydrates, and fat, respectively. A recommended composition of the standard breakfast is two eggs fried in butter, two strips of bacon, two slices of toast with butter, four ounces of hash brown potatoes, and eight ounces of whole milk. These meal conditions should provide the greatest effect on GI physiology, so that the systemic exposure of the drug is maximally affected. Depending on the clinical relevance of this maximum effect, specific recommendations on drug intake should be given [1]. For a long time, the FDA recommended only this type of test meal, but in February 2019 a new draft guidance was issued by the FDA, which recommended also testing the effect of a low-fat meal on oral bioavailability when drug intake with a high-fat meal is critical with regard to safety and efficacy [2]. Compared to a high-fat meal, a low-fat meal (providing a total of 400–500 kcal, with 25% of the calories coming from fat) can have less or no impact on systemic exposure for some drugs. The recommendation of a low-fat meal in the new FDA Draft Guidance is a first step towards the investigation of alternative meal compositions beyond the high-fat meal.

The drug plasma concentration time profiles obtained after oral administration of drug products are the results of a complex interplay of various factors related to either the physiology/pathophysiology of the subject or patient, the properties of the drug, or the properties of the formulation. PK data viewed in isolation rarely allow us to draw mechanistic insights into the processes of drug release and absorption. For these questions, further in vivo techniques can be applied to characterize the in vivo behavior of oral drug products, through the measurement of the dosage form’s location in the GI tract, the local conditions of fluid volume, pH, fluid composition, or mechanical stress, or through the measurement of drug regional permeability.

In principle, the application of in vivo methods in oral biopharmaceutics aims at the following questions:

1.
*How does an oral formulation travel through the human GI tract?*
In terms of the GI transit behavior of oral dosage forms, the size of the formulation as well as its disintegration and dissolution behavior are main factors;

2.
*How is the drug absorbed from different regions along the GI tract?*
The absorption (permeation) is the crucial biopharmaceutical parameter that determines whether the drug can be absorbed, or whether it is possible to formulate it into a modified-release formulation. Once the intestinal permeation properties are known, the decision of which formulation development program to use can be made;

3.
*What is the connection between fluid kinetics in the GI tract and oral drug absorption?*
The luminal fluid volumes present in the different parts of the human GI tract greatly affect the pharmacokinetic profiles of orally administered drugs. A deep knowledge of the fluid volumes available in the stomach and small intestine is one of the prerequisites for the successful development of oral dosage forms;

4.
*What kind of medium is available for the processes of disintegration and dissolution?*
Owing to the growing number of poorly water-soluble drugs entering the market, the physicochemical properties of the media present in the human GI tract (e.g., luminal pH value or bile salt concentration) are becoming increasingly important to understand luminal drug concentrations and to predict oral drug absorption.

This review, written by known experts in the field, aims to provide a practical guide to the different methods that can be applied to gain a deeper understanding of the in vivo behavior of oral drug products in the human GI tract. Our primary aim is to guide the reader through the jungle of in vivo methods, and to explain why and how these methods can be used to address certain questions around oral drug delivery. For this purpose, the individual sections follow the same structure, so that the same aspects are covered for each method. These include a brief description of the method, its potential applications in oral biopharmaceutics, including an example, as well as an overview on the advantages and limitations of each method. At the end of each section, an “Infobox” concisely summarizes the key messages. Finally, 3–5 key publications are recommended that can be used by the interested reader to dive further into the details of the method.

## 2. Physiological Factors Affecting Oral Drug Delivery

From a biopharmaceutical point of view, a thorough understanding of the most important parameters present in the human GI tract is required for the successful development as well as for the effective and safe application of oral drug products. This section will give a brief overview of human GI physiology and its potential effects on oral drug delivery. For a deeper description, the interested reader is referred to recent review articles [3,4,5].

The stomach is anatomically divided into the fundus, corpus, and antrum. These areas fulfil different physiological functions. While the fundus is primarily used to store ingested food, the antrum is used to crush the food and mix the stomach contents with secretions [6]. The corpus is the connective part between the fundus and the antrum. After oral administration of a solid dosage form, the release of the drug usually begins in the stomach. Thereby, the processes of disintegration and dissolution depend on various physiological parameters, such as the volume and the composition of luminal fluids, the physicochemical properties of the contents, and/or the prevailing motility pattern. It must be considered that these parameters are not static, but dynamic changes occur due to secretion, digestion, and emptying processes [7,8]. Finally, gastric emptying controls the rate at which the drug is transferred through the pylorus into the small intestine.

The small intestine consists of three regions: duodenum, jejunum, and ileum. These regions can differ in terms of the physicochemical properties of the contents as well as in their absorptive capacity. As the small intestine is usually the main site of drug absorption, the drug concentrations in the different regions of the small intestine are of great importance for oral drug delivery. The intestinal concentrations highly depend on the volume of luminal liquids and their physicochemical properties (e.g., pH, bile salt concentration). It is important to consider that the fluids are not distributed equally in the small intestine; rather, several fluid pockets of different sizes are predominant [9,10,11]. Simultaneous secretion, transfer, and absorption processes again lead to highly dynamic conditions. In the proximal part, the stomach continuously delivers mostly acidic chyme, which is then mixed with bile and pancreatic secretions. Here, the change in pH caused by the secretion of bicarbonate can have especially dramatic consequences on the concentration of the dissolved drug. In the case of weakly basic, poorly soluble drugs, precipitation can occur, and may limit drug absorption. However, the effect of precipitation may be mitigated by redissolution, the solubilization into micelles formed by the bile salts, and/or rapid drug absorption [3].

Although the colon is often not regarded as the main site of drug absorption, the conditions in the colon can also be relevant for oral drug delivery—especially for extended-release formulations as well as for colon-targeted formulations. The development of these formulations requires a thorough understanding of the rate of colonic absorption as well as the conditions for drug release [12].

## 3. Imaging Techniques Used in Oral Biopharmaceutics

Based on these physiological considerations, the gut can be regarded as being composed of a series of compartments through which the swallowed formulation—and its parts if it disintegrates—travels. Deducing the formulation’s residence in different fluid environments and transient mixing from the plasma concentration–time profile is a complex task, since without additional information, some phenomena cannot be distinguished. For example, a lag time in the PK data could be explained by a lack of drug release, lack of drug dissolution, or by gastric retention if the drug does not permeate the stomach wall. The area under the plasma concentration–time curve (AUC) is a summation of the pharmaceutical phase and the pharmacokinetic phase, as described by Ariens et al., and PK profiles sometimes display anomalies of missing peaks, multiple peaks, and unexplainable profiles [13].

The introduction to imaging was precipitated (literally) by a formulation of griseofulvin in PEG-400, which according to first principles should have been superior to the drug loosely packed in a capsule [14]. In fact, the opposite was true, and clearly the accelerated transit through the upper GI tract and the extent of dispersion were important factors that were initially ignored. This drove biopharmaceutical scientists to consider clinical imaging as an important tool, enabling the investigation of GI motility via a combination of electrophysiological measurements and imaging techniques.

Medical physics colleagues first studied GI motility in animals. Later, studies were also performed in humans with the newly emerging technology of gamma scintigraphy (see Section 3.1) [15,16]. In studies of oral absorption of drugs from test formulations, gamma scintigraphy is today often combined with PK measurements, to inform researchers as to the time and location of a dosage form’s disintegration (Figure 1). In particular, several studies performed in the 1980s and 1990s revealed marked differences between the behavior of an oral monolith versus pellets or dispersed forms in the presence of food [17]. Gamma scintigraphy was able to quantify the effects of formulation variables such as density, viscosity, and coating on the transit of oral drug formulations. Most importantly, the knowledge helped with dosage form optimization [18].

A weakness of the techniques in imaging studies is that anatomical identification is only moderate, and enthusiastic researchers may make claims that suggest unfeasibly fast transit due to confusion between the anatomy of the ileum and the colon. In a few countries, radionuclide-based studies are not allowed—especially regarding women of childbearing potential—stimulating research into alternative screening methods such as MMM or magnetic resonance imaging (MRI).

Magnetic resonance imaging became available to researchers throughout the late 1990s and beyond. Usually, images were acquired with the aid of research-minded medical physicists outside of their working hours. The key advantage of this technique is its exquisite definition, but the range of materials available as substrates is restricted. Therefore, soft gels have become increasingly interesting. Moreover, the imaging of oil fills works satisfactorily, as shown in Figure 2. At first, the movement of the stomach proved a problem, but it was found that turning the subject onto their stomach immobilized the gut contents. This was an old observation that had been noted by A.D. Keet, whose seminal work on radiological imaging of the upper gut from the beginning to the end of the last century in South Africa has been reprinted [20].

At the same time as MRI studies were performed by Clive Wilson et al. at Nottingham and Glasgow, Werner Weitschies was working in Berlin, and produced elegant MMM studies on the transit of iron-oxide-labelled tablets, showing the movement phases in the upper and lower GI tract (see Section 3.3). The research suggested that a useful return to “drill and fill” might even be possible. However, the measurements required a special facility in Berlin with the absence of anything remotely magnetic, which was too expensive to replicate. Fortunately, an alternative portable MMM system was developed, which could be used in the laboratory to follow the movement of a tablet containing iron oxide. There are relatively few of these machines (probably only two commercial systems were ever made), and the software was never fully developed beyond a simple prototype. The system was much less sensitive, demanding more iron oxide in each formulation, but nevertheless was able to follow the transit and disintegration of a tablet in vivo, and could certainly be developed further to investigate enteric and timed-release coatings in vivo. In a direct comparison with gamma scintigraphy, the instrument showed similar useful information [21].

### 3.1. Scintigraphy

#### 3.1.1. Application

Nuclear imaging is an established clinical imaging modality for monitoring metabolic processes in vivo, and has become increasingly important in drug product development and clinical pharmacology. Using techniques adapted from clinical radiopharmacy and nuclear medicine, drug molecules and formulations or carrier systems can be radiolabeled and their release, biodistribution, and uptake can be visualized in human subjects [22]. For instance, the technique of gamma scintigraphy has been identified as the preferred method to monitor the esophageal transit of a formulation.

#### 3.1.2. Description of the Method

For scintigraphic studies with drug products, the drug itself is not normally labelled, as the gamma-emitting radionuclides of the elements found in drugs generally have very short physical half-lives. The radiotracer is instead incorporated into various matrices, including liquids, gases, and solids. Clinical gamma camera imaging has relied heavily on Tc-99m (half-life 6.03 h), which is available from a clinical Mo-99 generator. Other radionuclides of different energies may be used in combination, permitting the examination of more than one phase of interest. For example, a component mixed with a meal may be labelled with technetium-99m (140 keV), whilst the formulation is labelled with In-111. The higher energy peak (247 keV) associated with In-111 is measured in a separate energy window, and the overlap is calculated. Similar experiments can be conducted with other pairs of radionuclides with appropriate energy filtering.

When undertaking scintigraphic studies of drug delivery, it is important that all procedures be carried out with full ethical and regulatory approval, and that manipulation of dosage forms is carried out to appropriate standards of good manufacturing practice (GMP). Any modifications to the formulation during the radiolabeling process must not affect the nature or in vivo behavior of the product [23]. For studies carried out to assess oral and GI transit, a non-absorbable radiolabeled tracer must be used for imaging. A number of different radionuclides may be incorporated into oral dosage forms such as liquid suspensions, tablets, and capsules. Both tablets and capsules can be radiolabeled by mixing finely dispersed ion-exchange resins and powders into the fill. If a research nuclear reactor is available, the complete dosage form can be irradiated, producing an activated marker with suitable characteristics for detection.

Tablets can be drilled with a small drill bit and filled with a suitable dry tracer. Once filled, the hole can be sealed with a suitable material such as an orthopedic adhesive. Careful examination of the surface should be undertaken to ensure that the nature of the surface coating has not been adversely affected. The following points should be considered for scintigraphic imaging of oral dosage forms:For gamma scintigraphy, the energy of the photons should be suitable for detection by a gamma camera—typically 40–400 keV—and the physical half-life must be sufficiently long for radiolabeling, dosing, and imaging to take place. Once the formulation has been tagged, the radiotracer should be bound stably, thus allowing the complete transit to be monitored;The most convenient radiopharmaceuticals for the gamma scintigraphy of oral drug delivery are Tc-99m and In-111. Typical examples include Tc-99m–diethylenetriaminepentaacetic acid (DTPA), Tc-99m–tin colloid, Tc-99m–sulfur colloid, and In-111–DTPA or ion-exchange resins;Neutron activation of the formulation addresses many the GMP issues of radiolabeling, since a non-radioactive form of the tracer may be incorporated prior to neutron activation in a research reactor. For example, the oxide of Sm-152 or Er-170 can be incorporated into the formulation during manufacture. Subsequent neutron irradiation in a reactor will activate the radioactive products Sm-153 and Er-171, respectively, which produce gamma energies suitable for imaging.

#### 3.1.3. Availability/Accessibility

As can be seen from the description above, this technique requires radiopharmacy facilities and nuclear medicine support. Therefore, such studies can only be conducted at specialized centers. In certain countries (e.g., Germany), scintigraphic studies cannot be performed with healthy volunteers because of the exposure to radiation.

#### 3.1.4. Example of Application

In a study, gelatin capsules were radiolabeled by the addition of Tc-99m ion-exchange resin and given to healthy volunteers. As shown in Figure 3, in one subject, the capsule was seen to remain in the lower esophagus for over 60 min, where it disintegrated, releasing its contents.

This example nicely illustrates how this technique allows in vivo observation of the swallowing of oral dosage forms. These methods have been especially valuable for the assessment of formulations for drugs that may cause esophageal ulceration or injury (e.g., bisphosphonates). They also have been used to study comparative performances and effects, such as the amount of water taken and the effect of patient posture at the time of swallowing [24].

Table 1 shows the advantages and limitations of scintigraphy in oral drug delivery research.
**INFOBOX**Nuclear medicine imaging can be used **for studying the transit, deposition, and release of oral drug formulations**. The **addition of a radiolabel** enables **in vivo imaging** and **quantification of drug deposition and dispersion**. These techniques are particularly valuable for the **study of the swallowing, GI transit**, and **site of release of oral dosage forms**.

#### 3.1.5. Recommended References

Frier, M.; Perkins, A. (Eds.) *Nuclear Medicine in Pharmaceutical Research*; CRC Press: Boca Raton, FL, USA, 1999. https://doi.org/10.1201/9781482272987.Perkins, A.C.; Blackshaw, P.E.; Hay, P.D.; Lawes, S.C.; Atherton, C.T.; Dansereau, R.J.; Wagner, L.K.; Schnell, D.J.; Spiller, R.C. Esophageal transit and in vivo disintegration of branded risedronate sodium tablets and two generic formulations of alendronic acid tablets: A single-center, single-blind, six-period crossover study in healthy female subjects. *Clinical Therapeutics*
**2008**, *30*, 834–844. https://doi.org/10.1016/j.clinthera.2008.04.018.Perkins, A.; Frier, M. Radionuclide Imaging in Drug Development. *Current Pharmaceutical Design*
**2005**, *10*, 2907–2921. https://doi.org/10.2174/1381612043383476.

### 3.2. Magnetic Resonance Imaging

#### 3.2.1. Application

MRI has started to provide biorelevant insight into the environment in which oral drug products are delivered and dissolve. A common application of MRI is the study of gastric emptying, whether of a standard dosing volume of water in the fasted state or the gastric emptying of standard test meals. Recently, this technique has also been used to study the intragastric fate of formulations and delivery devices using a variety of materials to label the dosage forms by means of altering the MRI signal provided by the dosage form. These vary from complex fluorinated compounds to simple materials such as lipids and pineapple pieces. Additionally, this technique provided the first demonstration that freely mobile water volumes in the small bowel are relatively small and distributed in pockets that can be quantified [9,25]. GI motility can be captured using dynamic cine MRI, and colonic volumes can be assessed as well.

#### 3.2.2. Description of the Method

MRI is a widely diffused medical imaging technique. The basic principles underpinning MRI are (1) the ability to transmit and receive radio signals to certain nuclei (but primarily the water hydrogen nuclei ^1^H) when they are placed in a strong magnetic field, and (2) the ability to spatially localize the signal received using additional magnetic fields. These two characteristics allow sampling of the inside of an animal or human body and reconstruction of cross-sectional images of good quality and excellent tissue contrast. This technique is used worldwide for radiological diagnostic purposes, but recently it has been shown to be able to provide unprecedented insights into GI motor function and luminal contents, without the need to disturb physiology with invasive procedures.

The principal advantages and limitations of MRI are summarized in Table 2.

#### 3.2.3. Example of Application

An example of a recent study is provided in Figure 4. The study investigated the presence of freely mobile water in the colon. In the figure, examples of anatomical and fluid-selective images and reconstructions of colon anatomy and colon water pockets are shown. These parameters are critical for the emerging field of colon-targeted formulations. MRI also has the potential to become the modality of choice for early-phase assessment of the functionality of drug products, confirming their proof of concept and helping to explain their modes of action. In addition, its lack of invasiveness and ionizing radiation makes it particularly suitable for repeated or prolonged studies, and for investigating gut physiology in children.

#### 3.2.4. Accessibility/Availability

MRI scanners are deployed primarily in tens of thousands of hospital settings worldwide, and a more limited number in specialized research laboratories, which can have also animal scanners. Therefore, this technique can be available, although it requires experienced operators and data analysis/interpretation specialists.
**INFOBOX**Magnetic resonance imaging (MRI) **is a non-invasive and ionizing-radiation-free tool** to **investigate the GI motor function** and assess **the properties of luminal contents**. In this field, MRI is applied for the simultaneous acquisition of parameters such as **motility, transit, flow, and fluid distribution**, which have a profound impact on drug dissolution and absorption. This investigation can also help in increasing the in vivo relevance of in vitro models. Therefore, MRI is an emerging tool for pharmaceutical sciences.

#### 3.2.5. Recommended References

Alyami, J.; Spiller, R.C.; Marciani, L. Magnetic resonance imaging to evaluate gastrointestinal function. *Neurogastroenterology and Motility*
**2015**, *27*, 1687–1692. https://doi.org/10.1111/nmo.12726.Murray, K.; Hoad, C.L.; Mudie, D.M.; Wright, J.; Heissam, K.; Abrehart, N.; Pritchard, S.E.; Al Atwah, S.; Gowland, P.A.; Garnett, M.C.; et al. Magnetic Resonance Imaging Quantification of Fasted State Colonic Liquid Pockets in Healthy Humans. *Molecular Pharmaceutics*
**2017**, *14*, 2629–2638. https://doi.org/10.1021/acs.molpharmaceut.7b00095.Schiller, C.; Fröhlich, C.-P.; Giessmann, T.; Siegmund, W.; Mönnikes, H.; Hosten, N.; Weitschies, W. Intestinal fluid volumes and transit of dosage forms as assessed by magnetic resonance imaging. *Alimentary Pharmacology and Therapeutics*
**2005**, *22*, 971–979. https://doi.org/10.1111/j.1365-2036.2005.02683.x.

### 3.3. Magnetic Marker Monitoring

#### 3.3.1. Application

MMM can be used for real-time tracking of the movement of dosage forms in the GI tract. When black iron oxide is used as a marker, the disintegration of the dosage form or the release of black iron oxide can be determined as a reduction in the magnetic moment of the dipole field. If there is a correlation between the release of a contained active substance and the decrease in the dipole field—for example, if the dosage form is eroding—the kinetics of the release of active substances from drug forms in vivo can also be quantitatively determined in this way.

#### 3.3.2. Description of the Method

This technique is also called magnetic pill tracking, gastrointestinal magnetomarkergraphy, or magnetic moment imaging. It is based on the labelling of dosage forms as a magnetic dipole. Since magnetic dipole fields are almost unaffected by non-magnetic materials such as body tissue, after ingestion of the labelled dosage form, the magnetic dipole field can be measured outside the body using suitable magnetic field sensors. Before ingestion, the magnetically labeled dosage form must be magnetized so that it generates a magnetic dipole field. From the measurement data, the strength (magnetic moment), orientation, and spatial position of the dipole can be calculated using special algorithms. The measurements are carried out with a high measuring frequency so that the spatial position of the dipole does not change during the measurement. Accordingly, with MMM, it is possible to measure positional data with very high temporal resolution.

In Table 3, the advantages and limitations of MMM are summarized.

#### 3.3.3. Example

In Figure 5, the movement of a magnetically labelled non-disintegrating capsule, which has been investigated by the utilization of MMM, is shown.

#### 3.3.4. Availability/Accessibility

The requirements for the measuring equipment are determined by the strength of the measuring signal, i.e., the dipole field. If strong magnets are used, standard gastric field probes are sufficient, which can also be operated in a non-magnetically-shielded environment by use of mobile sensor systems, which are commercially available. For highly sensitive biomagnetic measurement technology, a magnetically shielded environment is required. For this type of study, special centers with magnetically shielded rooms (e.g., PTB Berlin) must be involved. Here, pharmaceutical dosage forms can be labelled with very small amounts (1–10 mg) of ferromagnetic black iron oxide (E 170), which is commonly used as a pigment for coloring medicines and food products.
**INFOBOX**Magnetic marker monitoring (MMM) is a **non-invasive and radiation-free imaging tool** for the investigation of the **behavior (e.g., transit, disintegration, drug release) of solid oral dosage forms** such as tablets, capsules, or single pellets in the GI tract. It is based on the labelling of dosage forms as a **magnetic dipole** using ferromagnetic materials, as well as measurement of the dipole field distribution using arrays of magnetic field sensors. 

#### 3.3.5. Recommended References

Weitschies, W.; Kosch, O.; Mönnikes, H.; Trahms, L. Magnetic Marker Monitoring: An application of biomagnetic measurement instrumentation and principles for the determination of the gastrointestinal behavior of magnetically marked solid dosage forms. *Advanced Drug Delivery Reviews*
**2005**, *57*, 1210–1222. https://doi.org/10.1016/j.addr.2005.01.025.Weitschies, W.; Blume, H.; Mönnikes, H. Magnetic Marker Monitoring: High resolution real-time tracking of oral solid dosage forms in the gastrointestinal tract. *European Journal of Pharmaceutics and Biopharmaceutics*
**2010**, *74*, 93–101. https://doi.org/10.1016/j.ejpb.2009.07.007.Weitschies, W.; Wilson, C.G. In vivo imaging of drug delivery systems in the gastrointestinal tract. *International Journal of Pharmaceutics*
**2011**, *417*, 216–226. https://doi.org/10.1016/j.ijpharm.2011.07.031.

## 4. Invasive Approaches for Concentration Profiling and Permeability Assessments in Humans

The imaging techniques described in the previous section are mainly applied to study the transit and dispersion behavior of oral dosage forms in the GI tract. However, it is not possible (yet) to apply them for concentration profiling and permeability assessments. For this purpose, alternative methods such as aspiration of luminal contents or perfusion techniques are used.

### 4.1. Aspiration of Luminal Contents

#### 4.1.1. Application

GI sampling is a relatively invasive and labor-intensive technique that has been used by several research groups to investigate the driving force(s) of intestinal absorption after oral drug intake. The broader application of this technique in the last 20 years has contributed significantly to a better understanding of the drug concentration profiles occurring in the lumen of the stomach or the small intestine after the administration of oral dosage forms. In various studies, the roles of drug and formulation properties on luminal drug concentration were explored and linked to systemic drug exposure. In recent years, the technique has also been used to address specific questions such as the impact of food or antacid administration on the in vivo performance of several drug products, the effect of supersaturation versus solubilization, or the influence of hydrodynamics in the gastric environment on intraluminal drug and formulation behavior.

In view of multiple, simultaneously ongoing processes—including disintegration, dissolution, supersaturation, and precipitation—and a highly dynamic intraluminal environment, GI concentration–time profiles can be difficult to interpret. Nevertheless, there was significant progress over the last few decades that ultimately resulted in a better understanding of intraluminal drug and formulation behavior in the upper GI tract. In addition, efforts were also undertaken to achieve a better understanding of drug disposition in the distal small intestine and the human colon [28,29,30]. Data from studies in which complementary techniques were combined with GI sampling have been promising, and will hopefully allow us to close the gaps in our knowledge of intraluminal drug and formulation behavior.

Apart from focusing on drug- and formulation-related questions, GI sampling has also been used to characterize the composition of luminal fluids in different prandial states, as well as from different types of patients. These data have formed the basis for the development of various biorelevant media.

#### 4.1.2. Description of the Method

In GI sampling studies aiming at the characterization of drug concentrations in the upper GI tract, single or multiple catheters are positioned in the stomach, duodenum, jejunum, or ileum—either through the nose or through the mouth—as can be seen in Figure 6. Today, catheters with multiple sampling points can also be used, enabling the site-specific collection of fluids with just one catheter. The position of the catheters is typically checked by fluoroscopy.

Simultaneous blood sampling allows the systemic exposure of the drug of interest to be linked with the intestinal concentration–time profiles. Alternatively, it is also possible to combine the GI sampling technique with manometry, which is also described in this review. For instance, the link between gastric motility and drug distribution in the upper GI tract can be explored by combining these two methods. The used high-resolution manometry catheters have 36 pressure sensors, enabling monitoring of regional pressure events. With this technique, a drug can be administered during a specific phase of the migrating motor complex—a cyclical pattern of GI contractile activity [31,32,33].

#### 4.1.3. Accessibility/Availability

Due to the invasive character of this method and the complex procedure, GI sampling studies are typically conducted at a hospital under the supervision of a gastroenterologist. Although the equipment needed to conduct these studies can be purchased commercially, some experience is needed, which explains why GI sampling is not widely used in drug development.

Table 4 shows the advantages and limitations of the described aspiration techniques.

#### 4.1.4. Example of Application

The effect of proton pump inhibitor (PPI) intake on the dissolution/precipitation of basic compounds was illustrated using the protease inhibitor indinavir, administered as a sulfate salt (Crixivan^®^) [34]. The results of this study can be found in Figure 7. Under fasted-state conditions, a nearly complete dissolution of indinavir was observed (90 ± 3%), resulting in a maximum degree of supersaturation of 6.5 ± 5.9 in the intestinal environment. Concomitant intake of the PPI esomeprazole resulted in neutral pH conditions, and was accompanied with a lower fraction dissolved (58 ± 24%) and a reduced maximum degree of supersaturation (3.1 ± 1.3). These data illustrate that the intake of a PPI results in a reduction in intraluminal indinavir concentrations in the stomach as well as in the small intestine. These observations suggest that the reported reduction in systemic exposure of indinavir after PPI intake can be attributed to a lower driving force for intestinal absorption.
**INFOBOX**The GI sampling technique is an invasive but relatively straightforward way to study the **intraluminal drug concentrations in the human GI tract**. Depending on the type of question, single or multiple catheters can be placed in the stomach and/or small intestine to sample fluids from these regions and analyze drug concentrations. In most cases, the GI sampling technique is used to assess **the influence of the drug and formulation properties on luminal drug concentrations**. If combined with additional methods or techniques (e.g., blood sampling, manometry, MRI), this information can also be linked to aspects such as systemic drug exposure or GI motility. Furthermore, GI sampling data are often used as **reference data for physiologically relevant in vitro and in silico models**.

#### 4.1.5. Recommended References

Augustijns, P.; Vertzoni, M.; Reppas, C.; Langguth, P.; Lennernäs, H.; Abrahamsson, B.; Hasler, W.L.; Baker, J.R.; Vanuytsel, T.; Tack, J.; et al. Unraveling the behavior of oral drug products inside the human gastrointestinal tract using the aspiration technique: History, methodology and applications. *European Journal of Pharmaceutical Sciences*
**2020**, *155*, 105517. https://doi.org/10.1016/10.1016/j.ejps.2020.105517.Boyd, B.J.; Bergström, C.A.S.; Vinarov, Z.; Kuentz, M.; Brouwers, J.; Augustijns, P.; Brandl, M.; Bernkop-Schnürch, A.; Shrestha, N.; Préat, V.; et al. Successful oral delivery of poorly water-soluble drugs both depends on the intraluminal behavior of drugs and of appropriate advanced drug delivery systems. *European Journal of Pharmaceutical Sciences*
**2019**, *137*, 104967. https://doi.org/10.1016/10.1016/j.ejps.2019.104967.Brouwers, J.; van den Abeele, J.; Augustijns, P. Unraveling the Fate of Oral Drug Products in the Human GI Tract, *AAPS Newsmagazine*, 2018. Available online: https://www.aapsnewsmagazine.org/articles/2018/dec18/cover-story-dec18 (accessed on 15 February 2021).Vertzoni, M.; Augustijns, P.; Grimm, M.; Koziolek, M.; Lemmens, G.; Parrott, N.; Pentafragka, C.; Reppas, C.; Rubbens, J.; van den Abeele, J.; et al. Impact of regional differences along the gastrointestinal tract of healthy adults on oral drug absorption: An UNGAP review. *European Journal of Pharmaceutical Sciences*
**2019**, *134*, 153–175. https://doi.org/10.1016/10.1016/j.ejps.2019.04.013.

### 4.2. Determination of Intestinal Permeability by Perfusion Methods

#### 4.2.1. Application

Human effective intestinal permeability (P*eff*), together with major biopharmaceutical parameters such as drug solubility, dissolution rate, and GI transit, strongly affects the in vivo rate and extent of absorption and bioavailability from orally administered dosage forms [35,36]. The human intestinal P*eff* is a directly determined absorption rate (cm/s) across the intestinal mucosa, which is composed of a surface of mucin with the underlying epithelium. The human intestinal P*eff* for passive transcellular absorption is a diffusion process across the complex apical epithelial membrane barrier into the intracellular compartment. Single-pass intestinal perfusion models, based on the disappearance rate of the perfused drug from the intestinal segment, represent the absorption into the enterocyte. Intracellularly located first-pass metabolism, mediated by CYP3A4 or di- and tripeptidases, is not present in the outer apical leaflet of the enterocyte, and is considered not to affect the calculation of the intestinal P*eff*. Intracellular metabolism is, of course, a crucial part of the intestinal first-pass extraction, and is a major PK factor for many drugs. Low-molecular and polar compounds will also permeate to different extents through the intercellular space (paracellular transport).

In vivo predictions of human intestinal rate and extent of absorption of drugs are based on high-quality biopharmaceutical data of intestinal P*eff*, and clinical trials using human GI tubes are one way forward. There exist different types of intestinal perfusion approaches that have been extensively used in humans since the 1950s [37,38]. A useful approach has been to determine intestinal P*eff* with the single-pass perfusion of specific small intestinal segments. In addition, our research group has had the objective to increase human in vivo P*eff* in all relevant drug-absorbing regions of the small and large human intestine, and subsequently developed and validated a novel data analysis approach to calculate P*eff*. This approach is based on the appearance rate of the study drug in plasma, which has been shown to be important for drugs with low and highly variable intestinal P*eff* [35,37,38,39]. In these experimental in vivo investigations, it is crucial to examine luminal and brush border metabolism, and to check for binding of the study drug to the tubes, as these processes are expected to influence the accuracy for in vivo intestinal P*eff* [35,40,41,42].

#### 4.2.2. Description of the Methods

In short, four human intestinal perfusion techniques have been extensively applied. These include the following:The two open-perfusion systems: double lumen (L) and triple lumen (Triple-L);One semi-open proximal balloon (Prox-B) method;The double-balloon approach (Loc-I-Gut) (Figure 8).

In the triple-lumen-tube intestinal perfusion approach, the perfusion medium and GI fluids are mixed in a “mixing segment”. A baseline sample is taken from the distal end of this mixing segment. The determination of intestinal P*eff* with this method is based on a sample harvested at the end of the test segment (20–30 cm distal to the mixing segment) [37,38,39,40]. It is recognized that the luminal composition of the perfused mixing segment changes, which makes it difficult to maintain similar absorption conditions both within and between experiments. The perfusion solution with the study drug is also expected to flow in both directions in the luminal segment. These limitations with proximal contamination might be reduced by applying a multi-lumen tube with an occluding balloon proximal to the test segment. There also is the possibility to have a system with a separate tube aboral to the occluding balloon that continuously drains—and prevents proximal contamination of—the single-pass perfused test segment. Such a tube system would reduce the proximal leakage and make intestinal drug absorption conditions stable. Thus, the human intestinal P*eff* would be determined during well-defined luminal conditions. However, both of these single-pass perfusion open and semi-open approaches have a low recovery of any non-absorbable volume marker, such as PEG 4000. These GI tubes also need rather high perfusion rates in the range of 5–20 mL/min [37,38,39,40]. On the other hand, one main advantage is that distal parts of the small intestine can be investigated, as reported by Gramatté et al. [41,42].

Single-pass perfusion of the proximal small intestine with a double-balloon approach (Loc-I-Gut) as well as a colorectal segment (Loc-I-Col^®^) has been extensively used at our GI laboratory at Uppsala University [35,43,44]. When using the Loc-I-Gut technique, a small intestinal segment (length: 10 cm) between two balloons is single-pass perfused (Figure 8). One of the advantages of this perfusion approach is the occlusion of a test segment between the two intraluminal balloons, which maintain constant absorption conditions, as there is minimal contamination by GI fluids in the test segment. In addition, the leakage from the segment over the balloons is small, as indicated by the complete recovery of the non-absorbable marker [35,43,44]. This control of the absorption conditions in the intestinal segment provides excellent in vivo conditions for investigations of the transport and metabolism of drugs, nutrients, and other compounds in the human intestine. The human jejunal permeability with this GI tube (Loc-I-Gut) has been one of the cornerstones in the development and establishment of the biopharmaceutical classification system (BCS) [35,36]. In addition, these jejunal P*eff* data also formed the basis for several in silico models, correlation with various cell monolayers and preclinical absorption methods [45].

Human jejunal P*eff* has been extensively investigated by applying this single-pass, double-balloon approach in the proximal human small intestine during the last three decades [35,37,38,39]. Human intestinal P*eff* has been calculated from historical human perfusion data, which were based on open- or semi-open-perfusion GI systems, and experimental studies with GI tubes dosing directly into the lumen [37,38,39]. Intestinal P*eff* values from different clinical perfusion methods have been shown to be comparable [37,38,39]. It is also clear that the intra- and interindividual variability for low-permeability drugs was reduced when calculated from their appearance in systemic compartments [37,38,39]. This is explained by the fact that that any calculations based on drug disappearance from the perfused segment will be sensitive to small differences in the fraction absorbed during a single-pass perfusion.

#### 4.2.3. Example of Application

The intestinal P*eff* has seldom been determined in all parts of the human intestinal tract. By applying a GI tube–capsule technique (Bioperm AB, Lund, Sweden), small volumes under study can be delivered to predefined positions at different intestinal sites. The regional intestinal P*eff* was calculated by using an adopted intestinal deconvolution method [45]. The jejunal P*eff* values of atenolol, metoprolol, and ketoprofen were strongly correlated with the directly determined jejunal P*eff* values of the same drugs based on data from a single-pass perfusion experiment. These novel regional intestinal P*eff* values in humans have been useful in the evaluation and validation of preclinical biopharmaceutical tools, including their implementation in mechanistic in silico absorption models.

Table 5 summarizes the major advantages and limitations of intestinal perfusion tubes.
**INFOBOX****Human effective intestinal permeability** (*Peff*), is a major **biopharmaceutical variable**, and together with luminal drug stability, drug luminal solubility, luminal dissolution rate, and GI transit rate, it **strongly influences the in vivo rate and extent of intestinal drug absorption in both humans and animals**. Direct experimental in vivo approaches in humans and animals to examine intestinal drug absorption, secretion, and metabolism are feasible by applying **regional intestinal perfusion techniques**. In general, four different types of clinical GI tube have been employed in the small and large intestine to various extents. A significant correlation has been established between in vivo intestinal P*eff* and pharmacokinetically derived values for the fraction of dose absorbed for structurally different drug molecules.

#### 4.2.4. Recommended References

Dahlgren, D.; Roos, C.; Sjögren, E.; Lennernäs, H. Direct in Vivo Human Intestinal Permeability (Peff) Determined with Different Clinical Perfusion and Intubation Methods. *Journal of Pharmaceutical Sciences*
**2015**, *104*, 2702–2726. https://doi.org/10.1002/jps.24258.Dahlgren, D.; Roos, C.; Peters, K.; Lundqvist, A.; Tannergren, C.; Sjögren, E.; Sjöblom, M.; Lennernäs, H. Evaluation of drug permeability calculation based on luminal disappearance and plasma appearance in the rat single-pass intestinal perfusion model. *European Journal of Pharmaceutics and Biopharmaceutics*
**2019**, *142*, 31–37. https://doi.org/10.1016/j.ejpb.2019.06.011.Gramatté, T. Griseofulvin absorption from different sites in the human small intestine. *Biopharmaceutics and Drug Disposition*
**1994**, *15*, 747–759. https://doi.org/10.1002/bdd.2510150903.Lennernäs, H.; Ahrenstedt, Ö.; Hällgren, R.; Knutson, L.; Ryde, M.; Paalzow, L.K. Regional Jejunal Perfusion, a New in Vivo Approach to Study Oral Drug Absorption in Man. *Pharm. Res.*
**1992**, *9*, 1243–1251. https://doi.org/10.1023/A:1015888813741.Lennernäs, H. Human intestinal permeability. *Journal of Pharmaceutical Sciences*
**1998**, *87*, 403–410. https://doi.org/10.1021/js970332a.

## 5. Functional Diagnostic Tests Used to Describe the Physiological Conditions in the Human GI Tract with Respect to Oral Drug Delivery

Various diagnostic tests are applied in medical routine to assess specific functions of human GI physiology (e.g., gastric emptying, motility) in certain patient populations. In recent years, biopharmaceutical scientists have used these techniques to explore human GI physiology in healthy adults—who represent the subjects of many PK studies—as well as in certain patient populations. Thus, an improved understanding of human GI physiology was obtained, which resulted in optimized formulations as well as optimized in vitro and in silico tools for the prediction of drug and formulation behavior in the human GI tract.

### 5.1. Telemetric Capsules

#### 5.1.1. Application

Telemetric capsules can be used to explore the intraluminal conditions in the human and animal GI tracts in a minimally invasive manner. The most prominent system—the SmartPill™ GI monitoring system (Medtronic plc, Dublin, Ireland)—allows the measurement of pH, pressure, and temperature in high temporal resolution during GI transit. Based on this information, it is possible to determine transit times through the different regions (i.e., stomach, small intestine, colon) of the GI tract. Apart from the SmartPill™, further telemetric capsules are available [46]; their capabilities are summarized in Table 6.

In the past, telemetric systems have primarily been used to describe the transit times and conditions (especially pH) in different prandial states in humans, but also in different preclinical species. Their main advantage is that they can move freely within the GI tract and, thus, their transit behavior is regarded as being comparable to conventional oral dosage forms.

The information gathered by telemetric capsules can be used to gain a mechanistic understanding of drug release and absorption—especially if the capsule is applied in the frame of a pharmacokinetic study. Furthermore, the (patho)physiological information obtained from studies with telemetric capsules can also serve as inputs for biorelevant in vitro test methods, as well as for PBPK models.

#### 5.1.2. Description of the Method

Telemetric capsules are comparable in size to larger capsules (size 00-000), and typically contain one or more sensors to measure parameters such as pH, pressure, or temperature. The recorded data are regularly transferred to a data receiver, which must be worn close to the body of the subject. At the end of the experiment, the data can be transferred to a computer and evaluated with the corresponding software.

The IntelliCap^®^ system, which is currently no longer available, also had a drug release unit that allowed the selective and remote dosing of solid, semi-solid, or liquid formulations. The system was therefore mainly used to investigate region-specific absorption (e.g., from the colon) [47].

Further telemetric capsule systems with additional sensors are currently being developed in this research area. However, these systems are not yet commercially available.

In Table 7, the advantages and limitations of telemetric capsule systems presented.

#### 5.1.3. Accessibility/Availability

The application of telemetric capsules is relatively simple, and does not require special training. However, data analysis and interpretation can be difficult for inexperienced users, especially in terms of transit time determinations. Therefore, it is recommended that transit times are evaluated independently by at least two different observers.

As already mentioned, various telemetric capsule systems are commercially available [46]. Apart from the telemetric capsule itself, such a system typically consists of the following items: a receiver, a docking station, and a computer with the corresponding software. These systems are affordable, but it should be noted that the telemetric capsules are typically single-use systems, and the prices for a single capsule are relatively high.

#### 5.1.4. Example of Application

An exemplary graph can be seen in Figure 9. In this study, the conditions in the human GI tract after consumption of the high-fat, high-caloric meal recommended by the FDA for studies of food effects and fed bioequivalence were studied with the help of a telemetric motility capsule [48]. The aim of the study was to describe the transit conditions of oral dosage forms in a fed state with respect to transit times, pH, pressure, and temperature. It could be seen that the human stomach is characterized by highly dynamic conditions in terms of pH and pressure. These data were later used to mechanistically describe clinical data, but also as an input for biorelevant in vitro tools such as the Dissolution StressTest device or the GastroDuo [49,50].
**INFOBOX**Telemetric capsules represent a minimally invasive and relatively simple way to study the **physiological conditions in the human GI tract**. Different systems—such as the Heidelberg pH capsule or SmartPill™—which sometimes differ considerably in their functionalities, are commercially available. In most cases, the data from these systems are used to describe the **magnitude and variability of certain physiological factors** (e.g., pH, temperature, pressure), and to explain certain drug product performance phenomena.

#### 5.1.5. Recommended References

Dressman, J.B.; Berardi, R.R.; Dermentzoglou, L.C.; Russell, T.L.; Schmaltz, S.P.; Barnett, J.L.; Jarvenpaa, K.M. Upper Gastrointestinal (GI) pH in Young, Healthy Men and Women. *Pharmaceutical Research*
**1990**, *7*, 756–761. https://doi.org/10.1023/A:1015827908309.Koziolek, M.; Schneider, F.; Grimm, M.; Modeβ, C.; Seekamp, A.; Roustom, T.; Siegmund, W.; Weitschies, W. Intragastric pH and pressure profiles after intake of the high-caloric, high-fat meal as used for food effect studies. *Journal of Controlled Release*
**2015**, *220*, 71–78. https://doi.org/10.1016/j.jconrel.2015.10.022Söderlind, E.; Abrahamsson, B.; Erlandsson, F.; Wanke, C.; Iordanov, V.; von Corswant, C. Validation of the IntelliCap^®^ system as a tool to evaluate extended release profiles in human GI tract using metoprolol as model drug. *Journal of Controlled Release*
**2015**, *217*, 300–307. https://doi.org/10.1016/j.jconrel.2015.09.024.Weitschies, W.; Müller, L.; Grimm, M.; Koziolek, M. Ingestible devices for studying the gastrointestinal physiology and their application in oral biopharmaceutics. *Advanced Drug Delivery Reviews*
**2021**, *176*, 113853. https://doi.org/10.1016/j.addr.2021.113853.

### 5.2. High-Resolution Manometry

#### 5.2.1. Application

The manometry allows the measurement of phasic motility in different parts of the GI tract, including the esophagus, stomach, duodenum, small bowel, and colon. High-resolution manometry can be and has been combined with aspiration techniques in the recent past to evaluate the influence of gut motility on the behavior of oral drugs (see Section 4.1). Water-perfused manometry also has been combined with MRI to study the influence of gut motor and secretory responses to the ingestion of the quantity of water normally used in human clinical trials.

#### 5.2.2. Description of the Method

Manometry consists of the introduction of catheters with pressure sensors inside the gut lumen, which are able to measure the pressure created by contractions of the gut wall. This is considered the gold standard to assess gut phasic motility (short-lasting contractions) as compared to gut tone (prolonged pressure increase) in vivo in humans. In Figure 10, a schematic representation of the manometry catheter positioned inside the left colon (left) and of the phasic and tonic motor activity (right) is depicted.

Two systems are normally used: (1) water-perfused catheters, and (2) solid-state catheters. In the first case, the pressure is measured as resistance to water flow at a constant rate, which is transmitted as the pressure change to the strain gauges. The degree of resistance depends upon the amplitude and duration of the motor event. In the second case, pressure is measured by the strain gauge, where the application of pressure causes deformation of the diaphragm which, in turn, alters the resistance of the strain gauge. This upsets the balance of the bridge circuit, and results in current flow. Both systems are reliable, but in the first case it must be considered that water-perfused catheters introduce water into the gut lumen. This amount of water can affect the motility measurement in the event of prolonged measurements, as the accumulation of water can modify the viscosity of luminal content and induce distension of the gut wall, stimulating the contraction of the wall’s smooth muscles. Moreover, the introduced water can affect the evaluation of luminal contents during GI sampling. In contrast, solid-state catheters are normally more fragile and expensive [51].

In the past, the catheters were equipped with sensors spaced 5 cm and 10 cm apart in the case of upper and lower GI tract measurements, respectively (conventional manometry). Over the last 10 years, advances in the technology have allowed scholars to obtain catheters with sensors spaced 1 cm apart (high-resolution manometry), and this has improved the resolution of the gut motility measurement [52]. In Table 8, major advantages and limitations of this technique are reported.

#### 5.2.3. Example of Application

In Figure 11, the different colonic responses to drugs as measured by high-resolution can be seen.

#### 5.2.4. Accessibility/Availability

The application of manometry, along with its analysis and interpretation, requires special training. This is normally performed in hospital settings, as the intubation of the upper GI tract needs to be guided by fluoroscopy, while that of the colon is performed after endoscopy-based positioning. The water-perfused catheters are cheaper than the solid-state catheters.
**INFOBOX**High-resolution manometry enables the **study of the role of gut motility on the drug absorption/dissolution**. It is **invasive**, but is the only method that allows the **evaluation of different motor patterns** and their characteristics at the level of the **esophagus, stomach, small bowel, and colon**. 

#### 5.2.5. Recommended References

Brinck, C.E.; Mark, E.B.; Klinge, M.W.; Ejerskov, C.; Sutter, N.; Schlageter, V.; Scott, S.M.; Drewes, A.M.; Krogh, K. Magnetic tracking of gastrointestinal motility. *Physiological Measurement*
**2020**, *41*, 12TR01. https://doi.org/10.1088/1361-6579/abcd1e.Corsetti, M.; Costa, M.; Bassotti, G.; Bharucha, A.E.; Borrelli, O.; Dinning, P.; di Lorenzo, C.; Huizinga, J.; Jimenez, M.; Rao, S.; et al. First translational consensus on terminology and definitions of colonic motility in animals and humans studied by manometric and other techniques. *Nature Reviews Gastroenterology and Hepatology*
**2019**, *16*, 559–579. https://doi.org/10.1038/s41575-019-0167-1.Farré, R.; Tack, J. Food and symptom generation in functional gastrointestinal disorders: Physiological aspects. *American Journal of Gastroenterology*
**2013**, *108*, 698–706. https://doi.org/10.1038/ajg.2013.24.Kahrilas, P.J.; Bredenoord, A.J.; Carlson, D.A.; Pandolfino, J.E. Advances in Management of Esophageal Motility Disorders. *Clinical Gastroenterology and Hepatology*
**2018**, *16*, 1692–1700. https://doi.org/10.1016/j.cgh.2018.04.026.Scott, S.M. Manometric techniques for the evaluation of colonic motor activity: Current status. *Neurogastroenterology and Motility*
**2003**, *15*, 483–513. https://doi.org/10.1046/j.1365-2982.2003.00434.x.

### 5.3. Pharmacokinetic Markers

#### 5.3.1. Application

There are several methods that make use of the absorption characteristics of certain substances to evaluate physiological GI function or the behavior of dosage forms after ingestion. Usually, these methods are used to investigate various parameters of the GI functions, such as gastric emptying or orocecal transit time, but some of them can also be used to assess the in vivo behavior of oral dosage forms. In Table 9, the most relevant marker substances and their possible applications are summarized.

The combination of different marker substances is also possible. For example, by combining the paracetamol absorption test with the sulfamethizole capsule method, simultaneous measurements of liquid and solid gastric emptying can be achieved [54]. Since the sampling procedure of the marker methods requires no specialized equipment or instruments, such as an MRI scanner, they can be applied under more physiological conditions, or when other methods are not applicable—for example, during pregnancy or labor [55,56]. Conveniently, the use of marker substances does not alter the human GI physiology. Methods relying on pharmacokinetic markers allow only an indirect assessment of GI physiological function, inferred from their pharmacokinetic profiles, and pharmacokinetic interactions of co-administered drugs should be avoided in the marker choice through an extensive literature review.

On the one hand, the data gained from these methods are valuable for evaluating normal or abnormal GI behavior in patients. On the other hand, they may be implemented into physiologically relevant in vitro test methods and physiologically based pharmacokinetic (PBPK) models. Furthermore, a comparison of the in vivo characteristics of different dosage forms is possible.

#### 5.3.2. Description of the Methods

The paracetamol absorption technique (PAT) and the salivary tracer technique (STT) make use of the physicochemical properties and the absorption characteristics of the model drugs paracetamol and caffeine, respectively [57,58,59]. Both marker substances are regarded as BCS class I drugs. Due to their high solubility and high permeability, their plasma concentration kinetics are closely correlated with the kinetics of gastric emptying of fluids. Additionally, good correlations of saliva and plasma caffeine and paracetamol concentrations have been reported [60,61], making saliva a suitable PK matrix to sample instead of blood for these drugs.

After administration of the marker substance (dosing recommendations are summarized in Table 9), blood or saliva samples are taken and analyzed by liquid chromatography coupled with mass spectrometry (LC–MS). Usually, t_max_ and C_max_ are employed to evaluate the gastric emptying kinetics. The biggest downfall of saliva sampling is the likelihood of oral contamination, which can lead to false-high initial drug concentrations. This can be circumvented by the use of an ice capsule, which protects the oral cavity from the embedded drug substance. Another possible application of the STT is the determination of disintegration time of oral dosage forms by the onset of salivary caffeine concentrations [62]. For this procedure, caffeine has to be added to the dosage form. The first sample, in which caffeine is detectable, determines the disintegration time of the dosage form.

While paracetamol and caffeine are established pharmacokinetic markers for the emptying of liquids from the stomach, they are usually not used to assess the gastric emptying of solids or semi-solids. Sulfamethizole has been proposed as another pharmacokinetic marker that addresses this issue [63,64]. After the intake of a meal, 15 capsules filled with a sulfamethizole-containing matrix (see Table 9) should be swallowed by the subjects. Blood samples are then taken, and the sulfamethizole plasma concentration is measured as a surrogate parameter for the gastric emptying of digestible solids.

Whereas paracetamol, caffeine, and sulfamethizole are mainly used for exploring the physiology of proximal parts of the GI, sulfasalazine is a pharmacokinetic marker of colonic absorption [65,66]. Sulfasalazine consists of two moieties—sulfapyridine and mesalazine (5-aminosalicylic acid)—connected by an azo bond. Sulfasalazine is stable under the physiological conditions in the stomach and the small intestine. However, the microbiome abundant in the colon is able to split the azo bond of the molecule via an enzymatic reaction. The sulfapyridine is absorbed rapidly by the colon mucosa [67]. Both saliva and plasma concentration measurements of sulfapyridine are applicable for determining orocecal transit time and colonic arrival of dosage forms via the sulfasalazine/sulfapyridine method (SSM).

In Table 10, the advantages and limitations of pharmacokinetic markers are summarized.

#### 5.3.3. Accessibility/Availability

The use of pharmacokinetic marker substances is relatively easy, and requires no highly specialized equipment. If the pharmacokinetics are determined in blood plasma, trained personnel are needed for blood sampling. The PAT, the STT, and the SSM allow the use of saliva as a specimen. In these cases, the subjects can take samples non-invasively on their own, with very high temporal resolution, removing the need for a clinical environment.

For the sample preparation, adequate equipment and reagents are needed. The concentrations of the marker substance have to be determined by a validated analysis method. Usually, LC–MS analysis is the method of choice for the biological samples, and simultaneously the biggest cost factor.

#### 5.3.4. Example of Application

Figure 12 shows the results of a clinical study in which the disintegration time of conventional hard gelatin capsules was investigated. The capsules, containing 50 mg of caffeine, were administered in a fasted state with 240 mL of water. In order to compare the disintegration times of the capsules, saliva sampling and MRI scans were performed simultaneously. The comparison of the disintegration times revealed a difference of around 4 min between the two methods. Since capsule disintegration occurred mostly in the stomach, salivary caffeine concentrations only increased after disintegration was observed in MR images. This delay was also caused by gastric mixing and emptying. Nevertheless, the application of caffeine is still useful for characterizing the in vivo behavior of oral formulations and, thus, was used in follow-up studies.
**INFOBOX**By determining the pharmacokinetics of marker substances whose absorption characteristics correlate with GI functions, an **indirect observation of physiological GI conditions** is possible. **Paracetamol and caffeine** can be used for monitoring the **kinetics of the gastric emptying of liquids**. Additionally, caffeine is a feasible marker for the monitoring of **in vivo disintegration times** of orally applied dosage forms. For the monitoring of the **emptying of solids from the stomach**, the **sulfamethizole** capsule method has been proposed. **The sulfasalazine/sulfapyridine method** can be applied to determine **orocecal transit time**. All of these methods have in common the fact that they do not influence the GI physiology itself, no specialized equipment is needed, and they can be applied under more physiological conditions. 

#### 5.3.5. Recommended References

Sanaka, M.; Kuyama, Y.; Yamanaka, M. Guide for judicious use of the paracetamol absorption technique in a study of gastric emptying rate of liquids. *Journal of Gastroenterology*
**1998**, *33*, 758–791. https://doi.org/10.1007/s005350050177.Sager, M.; Grimm, M.; Jedamzik, P.; Merdivan, S.; Kromrey, M.-L; Hasan, M.; Koziolek, M.; Tzvetkov, M.V.; Weitschies, W. Combined Application of MRI and the Salivary Tracer Technique to Determine the in Vivo Disintegration Time of Immediate Release Formulation Administered to Healthy, Fasted Subjects. *Molecular Pharmaceutics*
**2019**, *16*, 1782–1786. https://doi.org/10.1021/acs.molpharmaceut.8b01320.Staniforth, D.H. Comparison of orocaecal transit times assessed by the lactulose/breath hydrogen and the sulphasalazine/sulphapyridine methods. *Gut*
**1998**, *30*, 978–982. https://doi.org/10.1136/gut.30.7.978.Asada, T.; Murakami, M.; Sako, Y.; Fukushima, Y.; Yonekura, Y.; Konishi, J.; Kita, T.; Miyake, T.; Asada, T. Sulfamethizole capsule method—A new method for assessing gastric emptying of solids. *Digestive Diseases and Sciences*
**1994**, *39*, 2056–2061. https://doi.org/10.1007/BF02088148.

## 6. The Benefits of a Deeper Understanding of Human GI Physiology: Perspectives from Different Fields

The information generated by the application of in vivo techniques can be used in different ways to support drug product development in academic and industrial settings. The following sections present how in vivo data can aid the development of biorelevant in vitro methods and powerful in silico models, as well as how the gathered data and knowledge can help in formulation development. Moreover, the value of the in vivo techniques with respect to nutrition is also highlighted.

### 6.1. Design and Optimization of Biorelevant In Vitro Methods

Biorelevant in vitro models can be powerful tools to understand the in vivo performance of oral dosage forms, and are thus of great importance in the early stages of drug development. However, to generate relevant data, reliable in vitro tools are needed. The development of such models is typically based on data generated by various methods from studies in healthy volunteers.

Essentially, biorelevant in vitro tools can be divided into two groups: One approach is to utilize relatively simple tools that are based on compendial methods in combination with biorelevant media. These media can be very different in their level of complexity, and range from aqueous buffer solutions to complex artificial media, reflecting the conditions inside the GI tract [68]. Such simple models can sometimes also use modified settings (e.g., stirring rates) to reflect GI physiological changes more realistically. The media composition can be adapted to reflect a particular GI compartment or typical prandial conditions [69]. The media composition is mainly based on in vivo data from aspiration studies. As discussed in Section 4.1, it is possible to collect and characterize GI fluids via nasogastric intubation. The resulting knowledge can subsequently be used for the development of biorelevant dissolution media. Today, a wide variety of dissolution media exist, covering different compartments and prandial conditions of the GI tract. In cases where drug solubility might be the limiting factor for absorption, the investigation of drug dissolution and/or solubility in different biorelevant media might be useful to estimate the in vivo performance of the drug product. For example, the weak base cinnarizine shows increased solubility in media simulating the fed intestine (fed-state simulated intestinal fluid, FeSSIF) compared with the fasted intestine (fasted-state simulated intestinal fluid, FaSSIF). This observation is in accordance with the in vivo performance, since cinnarizine shows a positive food effect [70]. Such investigations can also be very beneficial in the early developmental stages, since knowledge about drug solubility and dissolution rates can guide the choice of formulation strategies to overcome potential limitations and provide for more robust formulations against the various GI conditions.

Another approach for biorelevant dissolution testing is the use of complex models that simulate the complex and dynamic GI conditions. For instance, the influence of mechanical stresses as well as the dynamic changing conditions in the GI tract (e.g., pH, temperature) on drug dissolution can be investigated. The reliable simulation of physiological conditions requires extensive datasets. Due to the wide range of physiological aspects that must be considered, multiple in vivo methods are required to cover these aspects. The most relevant methods that are normally used for the implementation of in vivo data into in vitro tools are described in this section. For example, the contraction forces that are simulated by the dynamic gastric model are based on data from MRI studies [71]. MRI studies can also be used to determine the transit times of solid dosage forms through the GI tract, or to investigate the process of gastric emptying. MRI also provided the in vivo data for the simulated gastric emptying rates in the GastroDuo system [49,72]. During the development of this model, datasets from several in vivo methods were collected and evaluated. For this purpose, data from MRI investigations were used to establish realistic gastric emptying rates. Additionally, realistic pH and pressure profiles were implemented on the basis of data from telemetric capsules. Furthermore, data from MMM studies enabled the realistic simulation of movement speeds for the dosage forms to be integrated [72,73,74,75,76].

These are just a few examples to illustrate the relevance of in vivo investigations for the development of physiologically relevant in vitro tools. The utilization of in vivo methods can help to obtain deeper insights into physiological processes and, thus, may improve our understanding of the impact of these processes, which helps to refine the in vitro tools and associated test protocols, ultimately improving the mechanistic understanding of oral dosage form performance.

### 6.2. Design and Optimization of In Silico Methods

In silico modelling by physiologically based biopharmaceutics modelling (PBBM), in combination with biorelevant dissolution testing in simulated GI fluids, has become a well-established tool in pharmaceutical product development. It has provided great benefits by means of increased efficiency and reduced need for preclinical and human in vivo studies. In addition to supporting decision making in drug product development, there is also now an emerging use in the regulatory context to replace in vivo bioequivalence studies and justify drug substance particle size and drug product dissolution specifications in view of the anticipated drug performance in patients. The beauty of PBBM is that it integrates all aspects of the absorption process, and allows the modelling of complex relationships between drug release, dissolution, GI physiology, and drug intestinal permeability. Thereby, the physiological understanding obtained via the in vivo tools described in this publication has provided a key cornerstone for PBBM development and refinement. Initially, PBBM was focused on predicting average absorption data, but now there is an increasing interest in understanding the intra- and inter-subject variability in these parameters, as well as the impact of disease or excipients on the GI physiology and, thus, on PK. This has opened up the way for new approaches, where GI physiological tools are included in in vivo pharmacokinetic studies, forming the basis for a model of a specific drug. For example, in one study of a low-solubility basic drug, a telemetric capsule was included to capture the variation in gastric pH and gastric emptying. It could be shown in PBBM that this was a critical factor for the drug absorption in individuals [77]. In doing so, much more accurate modelling could be performed. This approach may be extended to other factors—such as bile concentrations, motility, and local fluid volumes. Another area for novel developments in the PBBM area, similar to simulated fluids, is an increasing interest in characterizing patient populations where GI conditions vary compared to healthy volunteers, thereby allowing for a more patient-centric product development.

### 6.3. Formulation Development

Understanding of the oral GI physiology by use of in vivo tools has been critical for industrial drug product development over the last 40 years. This has been especially pertinent in the areas of modified-release (MR) products, as well as for immediate-release (IR) formulations of low-solubility drugs. This has happened through direct use of in vivo tools in the development, but also—perhaps more often—indirectly through translation of the understanding of GI physiology into relevant in vivo predictive tools. This has both led to innovative products with improved clinical performance in patients, and also supported a more efficient development process, bringing important drugs to market faster. While MR products and low-solubility drugs are still very common in industrial research and development (R&D), there are also additional drivers for improved understanding and tools for GI physiology and absorption. One example is the renewed interest in oral delivery of complex, larger molecules such as peptides and oligonucleotides, now also enriched with additional new modalities, often being hybrid molecules conjugating different entities. As these molecules typically have no sufficient oral bioavailability, advanced drug delivery based on understanding the GI physiology is required to enable oral application. As described in the previous section, another area that been very well established in the industry during the last 20 years is the prediction of in silico absorption via physiologically based pharmacokinetic modelling (PBPK)—recently rebranded in this context as PBBM—by the use of advanced software tools (e.g., GastroPlus^®^, Simcyp^®^, PK-SIM^®^). The mentioned aspects of industrial importance of understanding of the GI physiology will be exemplified in some more detail in the current section.

In vivo imaging has been extensively used for MR product development in order to understand GI transit phenomena under different conditions, as well as in vivo drug release (e.g., matrix tablet erosion). The majority of human studies have been performed by gamma scintigraphy, but other techniques—such as MMM, MRI, and X-ray—have also been successfully used. A key general lesson obtained in the area of oral MR was the dependence on size for gastric emptying in the fed state. When the stomach digests food, larger particles are maintained until they are ground to a small enough size, or until a fasting state is recovered. This has had very important implications in the design of enteric-coated formulations. For single units, such as tablets or capsules, it was shown by in vivo imaging that they could remain for over 24 h due to frequent food intake, while smaller objects (<1 mm) emptied more consistently with food. Thus, it has been possible to ascertain the desired onset of action, as well as to avoid overdosing by the simultaneous release of multiple tablets. Another more general finding in the MR area was the detection of a physiological dose-dumping effect for controlled-release tablets. It was shown by MMM that under fed conditions, the drug was released from the matrix of a controlled release (CR) tablet in the upper stomach at a similar rate as in the fasted state. However, since the drug was poorly mixed with food content for a certain period, the released drug was accumulated in the stomach without being absorbed. Eventually, the whole amount of the released drug was rapidly emptied out of the stomach, leading to a peak in the plasma drug concentration. This was not caused by rapid release from the CR tablet, but by the physiological dose-dumping out of the stomach. Thus, this has offered insights for more rational designs of CR products.

A critical area in the development of IR formulations of low-solubility drugs as well as oral MR products is the possibility of predicting the dissolution in the GI tract. As described previously, the characterization of GI fluids through telemetric devices and fluid sampling has been the foundation for the establishment of a set of different simulated GI fluid media covering different regions of the GI tract, as well as different conditions—such as fasted or fed states. There is also now an increased interest in characterizing certain patient groups where GI physiology has been altered. Another area that has been less extensively addressed, but is still significant for industrial development, is the hydrodynamic forces prevalent in the GI tract. This is most important for hydrophilic gel matrix tablets, as well as other eroding or slowly disintegrating formulations. It may also be of importance for precipitations of supersaturated solutions, which could occur for low-solubility drugs. The measurement by telemetric capsules and other manometric techniques has provided the basis for the design of in vitro test apparatuses mimicking the conditions in the GI tract. One example is the advanced gastric compartment model developed for the artificial GI tract tool the TNO Gastro-Intestinal Model (TIM), developed for in vitro absorption studies, where the gastric motility is validated versus manometric data. In particular, the use of simulated GI fluids in dissolution has been a well-established standard for a long time. This is not surprising, since the benefit in terms of the reduced need for lengthy and costly in vivo studies by use of predictive in vitro methods has had a significant impact on reducing the time and cost of product development.

### 6.4. Nutritional Sciences

In nutritional sciences, the bioavailability of a nutrient is defined as the proportion of a nutrient in food utilized for normal body function [78]. In order to study the bioavailability of nutrients, the methodology of nutrikinetics can be used. Nutrikinetics is similar to pharmacokinetics, but extended to the inclusion of the food matrix and the phenotype. Both modulate the bioactivity of dietary components—especially when the gut microbiota are involved [79]. The importance of the phenotype and gut microbiota on metabolic markers such as blood glucose has been shown in a study by Zeevi et al. in 2015, indicating that the gut microbiota has a direct or indirect effect on the absorption of the dietary components [80]. Furthermore, the food matrix effects the start, rate, and extent of absorption of dietary compounds [79]. According to Fardet, the food matrix can be defined as “as the support, architecture, or the structure of the food resulting from nutrient interactions, giving to it its form, thickness, density, hardness, porosity, color, and crystallinity” [81]. It can interact with the gut microbiota and influence the bioavailability and bioconversion of other food compounds, but also of active pharmaceutical ingredients (APIs) [79]. Therefore, it can be assumed that diet can have a post-digestion effect on the absorption of nutrients and pharmaceutical compounds.

Although different in vivo techniques are commonly applied in the nutritional sciences, the use of telemetric capsules has not been very common in classical nutritional sciences. However, they can provide an added value, as there is a lack of information about the bioavailability of nutrients/foods in specific patient populations. Moreover, the effect of the food matrix can be measured using a telemetric capsule (SmartPill^TM^) to reflect the gastric emptying time (GET). Thus, it could be shown that a liquid breakfast was emptied from the stomach on average 1 h faster than an isocaloric and fiber-matched solid breakfast [82]. Further examples of the utilization of telemetric capsules include the investigation of the GET after consumption of a specific kiwifruit- or a FODMAP (fermentable oligo-, di-, and monosaccharides and polyols)-based diet [83,84]. None of these studies focused on the effects of the gut microbiota, and only represented acute studies. In a recent study on the effects of kiwifruit consumption on gut microbiota composition, the pH was measured by the use of a telemetric capsule. The consumption of the kiwifruit capsules led to a lower pH and, consequently, to an abundance of *Bifidobacterium* spp., known as an established probiotic genus [85]. In all of the abovementioned studies, the telemetric capsules were used to measure GET, pH, or pressure. However, none of the studies described how the metabolites produced by the gut microbiota affected the absorption of pharmaceutical compounds. Apart from the latter, and the specific time period of these intervention studies, the lack of a convenient and accurate method to measure the dietary exposure was remarkable. A proper method to measure diet/food/nutrient consumption is key to unraveling the chronic effect of food consumption on drug absorption and drug–metabolite interaction—the so-called post-digestion effect of food on drug absorption.

The current lack of evidence about the post-digestion effect of food (matrix) on the bioavailability of both nutrients and pharmaceutical components indicates that there is a need for further research into the underlying processes that occur in the stomach, colon, and other parts of the GI tract [86]. More insights about the effects of the food matrix could lead to insights into the unwanted burst effect (sudden increase in concentration (30–60%) of specific materials within a relatively short time period) [87]. Proper in vivo data are needed before dynamic in vitro models can be further developed. In vivo tools to capture the spatial and temporal variations in the different gastric characteristics (e.g., pH, pressure, temperature) would assist in a more adequate prediction of the fate of food and pharmaceutical compounds in the GI tract [88].

Insights into the GI physiology of individuals are highly relevant, as several pathologies could induce a reduction in the bioavailability of nutrients and medical drugs. Until now, the GI physiology in specific patient populations has not been sufficiently researched [89]. One specific patient population is patients who have undergone bariatric surgery. Among bariatric surgeries, Roux-en-Y gastric bypass (RYGB) is considered to be the gold standard, but sleeve gastrectomy (SG) is rapidly gaining ground [90]. Bariatric surgery alters the anatomical structure of the GI tract by reducing the size of the stomach (i.e., inducing restriction in absorption). RYGB bypasses the proximal part of the small intestine as a surplus (i.e., inducing malabsorption). Knowledge of the GI physiology of the patients who have undergone bariatric surgery is currently limited. Therefore, we started the investigation of the physiological changes in the upper GI tract (i.e., pH, temperature, gastric emptying, intestinal transit time, motility, activity, and concentration of GI enzymes, and electrolyte distribution) in vivo in patients one year after RYGB or SG. In order to characterize these changes, we used the SmartPill^TM^. As shown in Figure 13, the differences in pH, pressure, and temperature in a person with RYGB and a person with a SG, compared to those of a person with an intact GI tract (Figure 9) are apparent.

## 7. Conclusions

Today, there is a wide variety of in vivo methods to characterize the human GI physiology and to investigate the fate of oral dosage forms after ingestion. Depending on the question to be investigated, different methods are suitable. For some investigations, there is unavoidable interference with gut function, which may alter the normal digestive process. For example, the characterization of the composition of GI fluids is only possible by aspiration. In contrast, other parameters—such as the pH or pressure forces present in the GI tract—can also be measured by non-invasive methods, such as telemetric capsules.

The utilization of the discussed techniques has led to great improvements in our understanding of the behavior of oral dosage forms after ingestion, as well as that of the GI tract itself. Several research fields have benefitted from this progress. For instance, in silico and in vitro models have been developed based on data generated in vivo. Various models with different degrees of complexity have been developed and used. The application of these models in the field of pharmaceutical formulation development can reduce the need for expensive in vivo studies, improve mechanistic understanding of the components limiting absorption, and help to identify the sources of clinical variability. The expanded knowledge base facilitates the development of robust formulations that can deliver consistent performance in the target patient group. In addition, the techniques presented in this review can also be used when the in vivo performance of novel oral drug formulations is investigated. Their application can help to obtain a better understanding of local and dynamic conditions for drug release, dissolution, and absorption, and thus facilitate the interpretation of PK profiles.

Nevertheless, there are still many unanswered questions today regarding the physiology of the GI tract in health and disease. For example, the physiological conditions in individuals or specific patient groups, and the implementation of these data into in vitro and in silico models, are currently of interest. Additionally, the development of in vivo imaging techniques opens up new possibilities for the research of physiology and formulation development. For instance, newer MRI scanners are able to perform measurements in the sitting position, creating new opportunities for investigations of the (human) GI tract. The discussed in vivo methods have been and will continue to be of outstanding importance for oral biopharmaceutics.

## Figures and Tables

**Figure 1 pharmaceutics-14-00801-f001:**
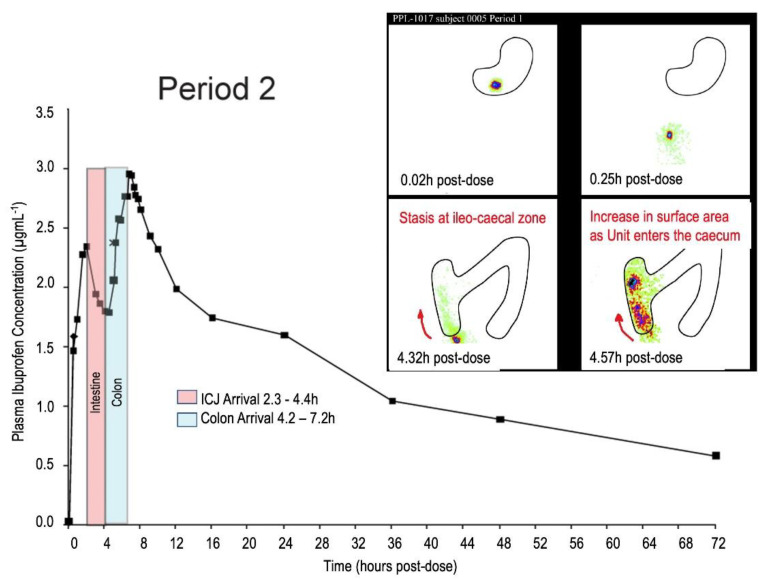
PK of an ibuprofen preparation in the cecum overlaid with intestinal and colonic localization. Colon arrival was associated with the second peak of drug concentration (ICJ: ileocecal junction) Based on *International Journal of Pharmaceutics*, 50, Wilson et al., “Bimodal release of ibuprofen in a sustained-release formulation: a scintigraphic and pharmacokinetic open study in healthy volunteers under different conditions of food intake”, 155–161. [19]. Copyright Elsevier (1989).

**Figure 2 pharmaceutics-14-00801-f002:**
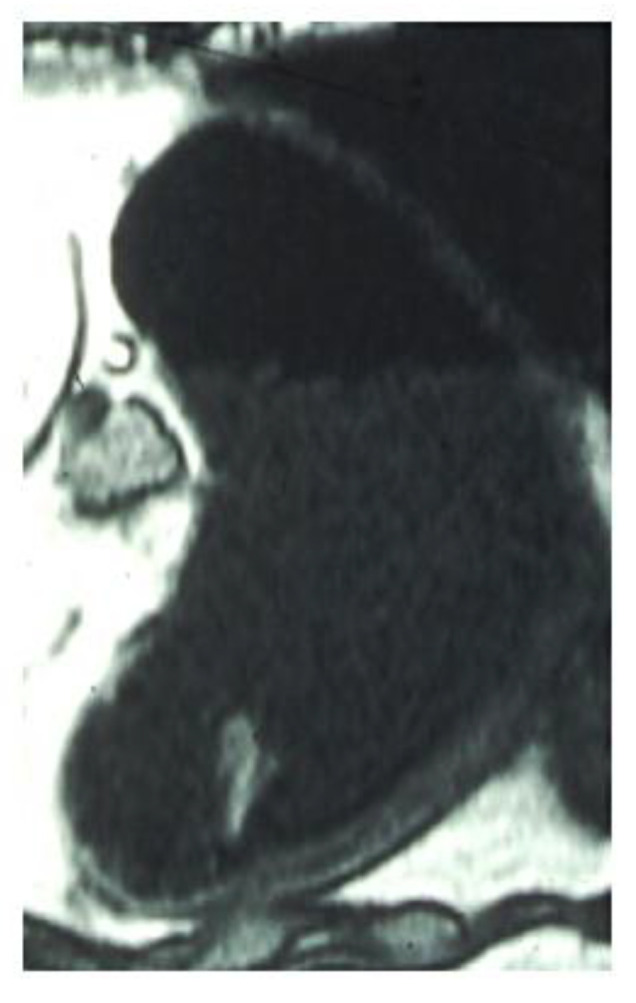
Oil-filled capsule disintegrating in the stomach.

**Figure 3 pharmaceutics-14-00801-f003:**
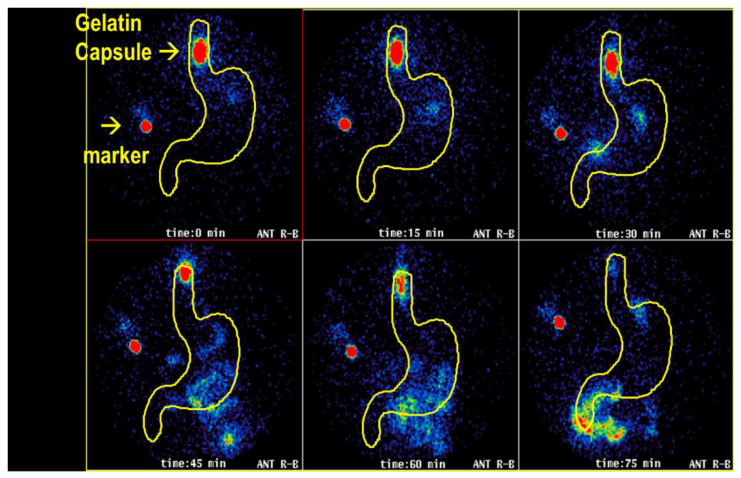
A series of gamma camera images showing a gelatin capsule formulation (shown in red pixels) stuck in the throat of a healthy volunteer. The subject was not aware that swallowing had not been completed for over 60 min.

**Figure 4 pharmaceutics-14-00801-f004:**
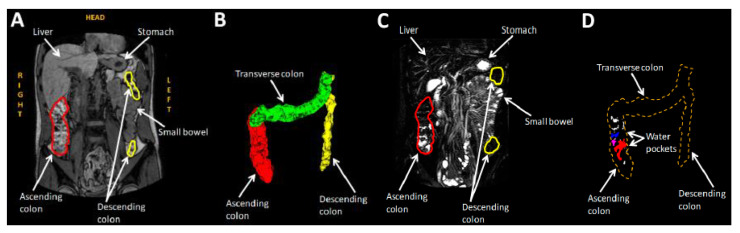
(**A**) Example of a coronal plane anatomical MRI abdominal image. Visible sections of the colon are highlighted. (**B**) Example 3D reconstruction of the colon. Red, green, and yellow differentiate the ascending, transverse, and descending colon, respectively. (**C**) Fluid-selective MRI image corresponding to (**A**); freely mobile liquid appears white, whereas less mobile fluid and tissues appear dark. (**D**) Extraction of freely mobile water pockets in the colon. Reproduced with permission from *Molecular Pharmaceutics*, 14, Murray et al., “Magnetic Resonance Imaging Quantification of Fasted State Colonic Liquid Pockets in Healthy Humans”, 2629–2638 [26] (https://pubs.acs.org/doi/10.1021/acs.molpharmaceut.7b00095, accessed on 21 February 2021), Copyright American Chemical Society (2017); further permissions related to the material excerpted should be directed to the ACS.

**Figure 5 pharmaceutics-14-00801-f005:**
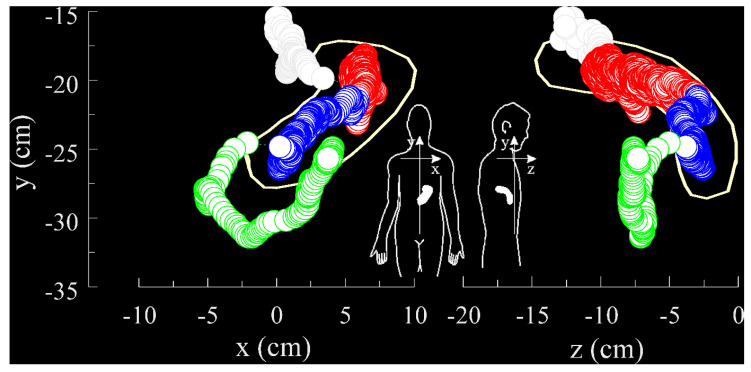
Example of the tracking of a magnetically labelled non-disintegrating capsule in the distal esophagus (grey circles), the proximal stomach (red circles), the distal stomach (blue circles), and the duodenum (green circles) by MMM. Every circle represents a localization step of 40 ms. Reproduced with permission from *International Journal of Pharmaceutics*, 417, Weitschies and Wilson “In vivo imaging of drug delivery systems in the gastrointestinal tract”, 216–226, [27], Copyright Elsevier (2011).

**Figure 6 pharmaceutics-14-00801-f006:**
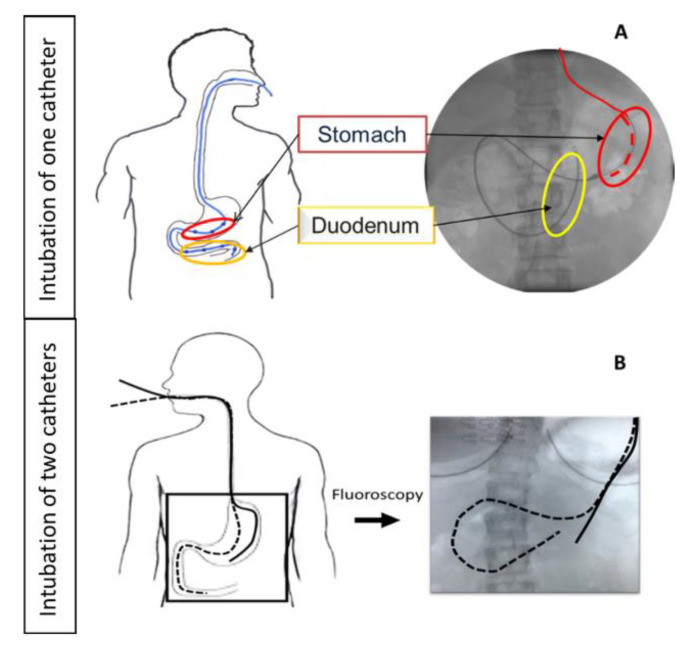
Schematic overview of the clinical aspiration technique—Left: Catheter(s) introduced through the nose (**A**) or mouth (**B**) and positioned inside the stomach and the upper small intestine. Right: X-ray images of (**A**) one catheter positioned inside the stomach and the intestine with aspiration/perfusion ports in both segments, and (**B**) a catheter positioned inside the stomach (solid line) and a catheter positioned inside the intestine (dotted line); catheters are traced for clarity. Adapted from *European Journal of Pharmaceutical Sciences*, 134, Vertzoni et al., “Impact of regional differences along the gastrointestinal tract of healthy adults on oral drug absorption: An UNGAP review”, 153–175, [3]. Copyright Elsevier (2019).

**Figure 7 pharmaceutics-14-00801-f007:**
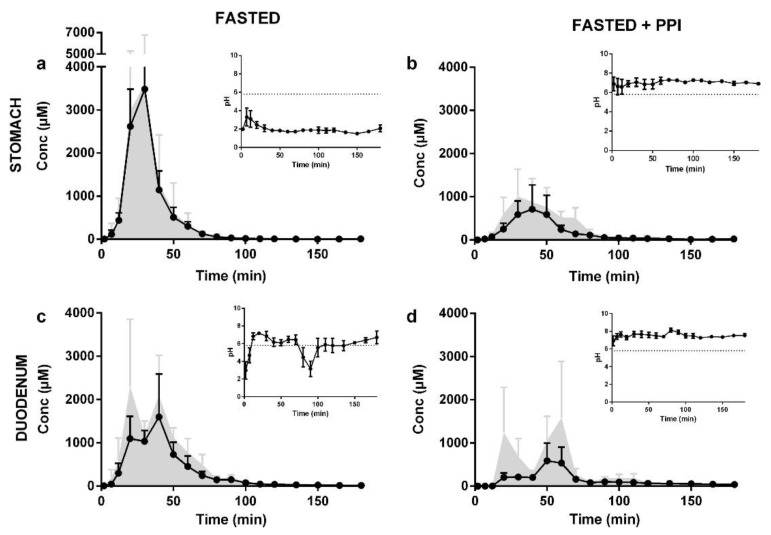
Gastric and duodenal concentration–time profiles for indinavir following administration of Crixivan in a fasted state (**a**,**c**) and a fasted state with concomitant use of the PPI esomeprazole (**b**,**d**). One capsule of Crixivan was orally ingested with 240 mL of tap water. Black lines and gray areas represent the dissolved and total indinavir content (solid + solute expressed as concentration), respectively. Inserts depict the corresponding pH profiles as a function of time. The dashed line represents the pKa of indinavir (5.8). (**a**) Fasted state, stomach; (**b**) fasted state with concomitant PPI use, stomach; (**c**) fasted state, duodenum; (**d**) fasted state with concomitant PPI use, duodenum (mean + S.E.M., *n* = 5). Reproduced with permission from *European Journal of Pharmaceutics and Biopharmaceutics*, 109, Rubbens et al. “Gastrointestinal dissolution, supersaturation and precipitation of the weak base indinavir in healthy volunteers”, 122–129, [34], Copyright Elsevier (2016).

**Figure 8 pharmaceutics-14-00801-f008:**
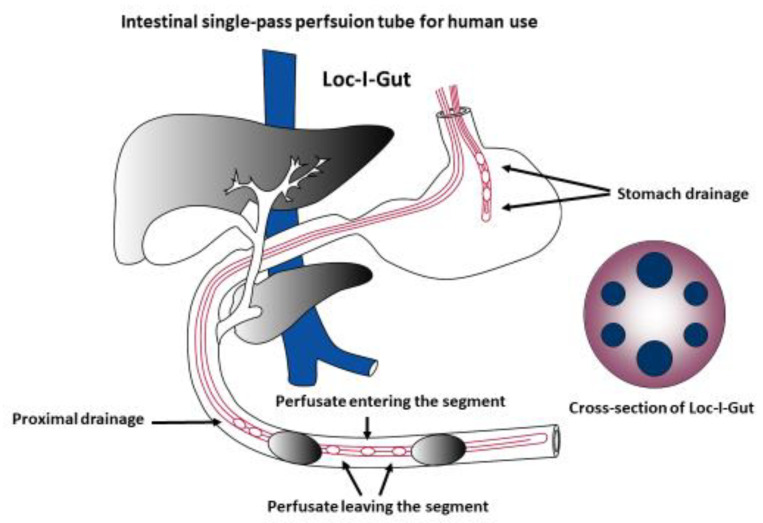
Schematics of the intestinal perfusion system used for in vivo studies in humans: Loc-I-Gut is a single-pass perfusion, double-balloon technique for the proximal small intestinal regions. The multichannel GI tube made from polyvinyl chloride (PVC) is 175 cm long, has an external diameter of 5.3 mm, contains six channels, and distally has two 40 mm long latex balloons (10 cm apart). These balloons are separately connected to one of the smaller channels, and the two wider channels are for infusion and aspiration of the perfusion solution. The two remaining smaller channels are for the infusion of marker substances and/or for drainage. A tungsten weight facilitates the passage of the tube into the jejunum. ^14^C-PEG 4000 is used as a non-absorbable volume marker to monitor water flux across the intestinal mucosa throughout the in vivo perfusion experiment.

**Figure 9 pharmaceutics-14-00801-f009:**
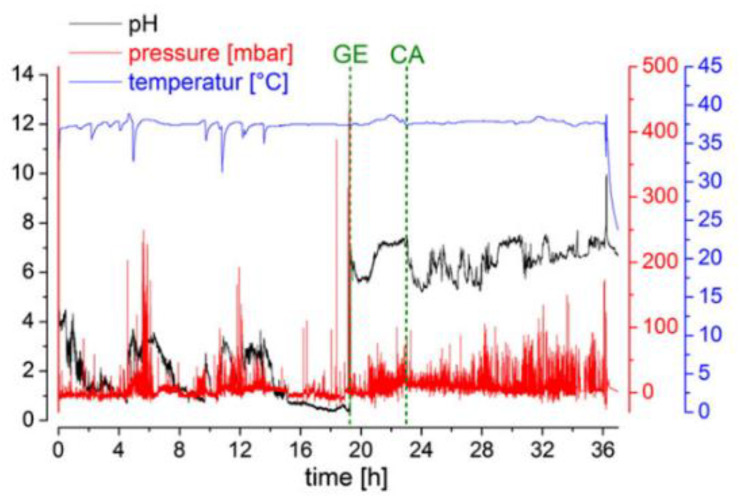
Exemplary pH (black), pressure (red), and temperature (blue) profiles over time obtained after SmartPill^TM^ administration in a fed state in a healthy volunteer (GE—gastric emptying, CA—colonic arrival). Reproduced with permission *from Journal of Controlled Release*, 220, Koziolek et al., “Intragastric pH and pressure profiles after intake of the high-caloric, high-fat meal as used for food effect studies”, 71–78, [49], Copyright Elsevier (2015).

**Figure 10 pharmaceutics-14-00801-f010:**
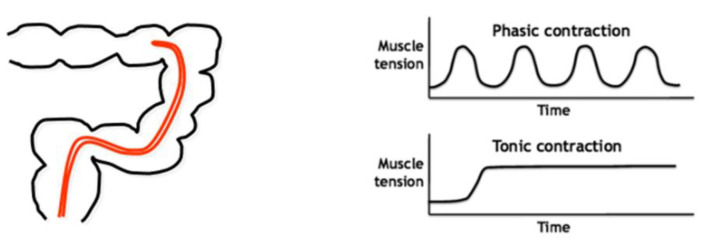
Schematic representation of the manometry catheter positioned inside the left colon (**left**), and of the phasic and tonic motor activity (**right**).

**Figure 11 pharmaceutics-14-00801-f011:**
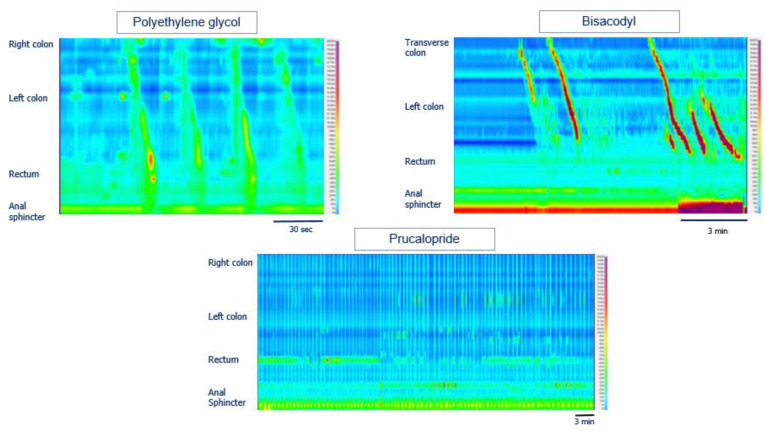
Motility patterns in the colon, the rectum, and the anal sphincter after exposition to polyethylene glycol, bisacodyl, and prucalopride, respectively. Reproduced with permission from *Neurogastroenterology and Motility*, Corsetti et al., “High-resolution manometry reveals different effect of polyethylene glycol, bisacodyl, and prucalopride on colonic motility in healthy subjects: An acute, open label, randomized, crossover, reader-blinded study with potential clinical implications” [53], © 2020 John Wiley & Sons Ltd.

**Figure 12 pharmaceutics-14-00801-f012:**
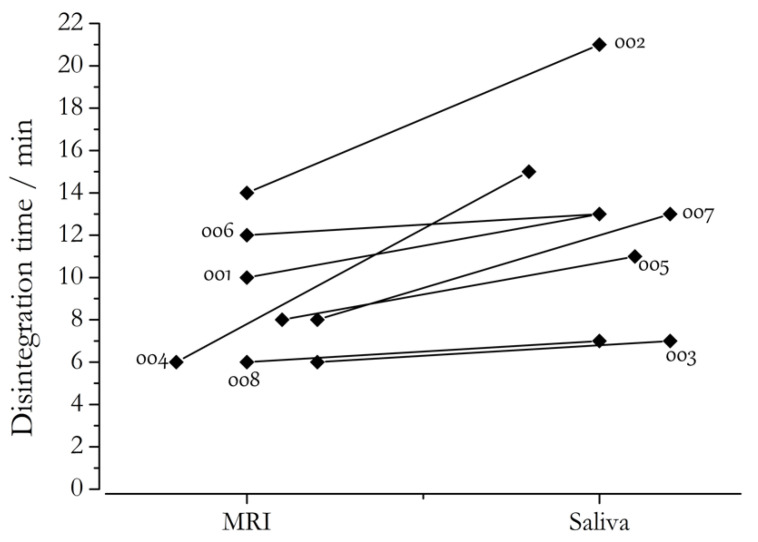
In vivo disintegration times of hard gelatin capsules that were labeled with 5 mg of black iron oxide and 50 mg of caffeine, as determined by MRI and the STT. Numbers indicate a single volunteer. Adapted from *Molecular Pharmaceutics*., 16, Sager et al., “Combined Application of MRI and the Salivary Tracer Technique to Determine the in Vivo Disintegration Time of Immediate Release Formulation Administered to Healthy, Fasted Subjects” 1782–1786 (2019) [62], Copyright American Chemical Society (2019).

**Figure 13 pharmaceutics-14-00801-f013:**
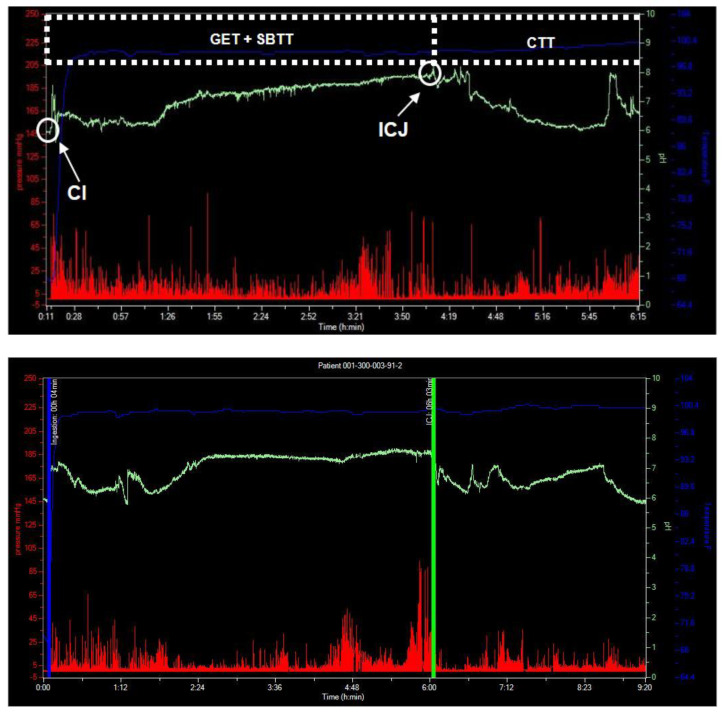
pH (green), temperature (blue), and pressure (red) curves in a fasted state after sleeve gastrectomy (SG, **above**) and Roux-en-Y gastric bypass (RYGB, **below**), measured by Smartpill^TM^. CI: capsule intake, GET: gastric emptying time, SBTT: small bowel transit time, ICJ: ileocecal junction, CTT: colon transit time.

**Table 1 pharmaceutics-14-00801-t001:** Advantages and limitations of scintigraphy.

Advantages	Limitations
Imaging of drug delivery without altering the physiological properties of the formation.	Uses small doses of radioactivity (normally less than those used in diagnostic imaging).
Enables the in vivo study of transit, release, and deposition throughout the entire GI tract, as well as clearance.	Requires radiopharmacy facilities and nuclear medicine support.
Suitable for monitoring rapid processes such as tablet swallowing.	Images provide functional information but lack anatomical information.
Allows quantification of kinetics and release, and can be used together with blood sampling for the assay of drug absorption.	Sensitivity and resolution of image data may restrict use for some applications.
	Positron emission tomography (PET) imaging may offer more information for direct labelling of drug molecules.

**Table 2 pharmaceutics-14-00801-t002:** Advantages and limitations of magnetic resonance imaging.

Advantages	Limitations
Non-invasive.	Unsuitable for patients with metal implants.
Use of non-ionizing radiation.	High instrumentation cost.
Good quality cross-sectional images.	Susceptible to motion.
Richness of soft tissue contrast.	Amount of data acquired can be a burden.
Acquisition of different parameters within a study session.	Lack of standardization and/or validation against gold-standard techniques.
New open configuration magnets can scan people in a sitting position.	Higher image distortion and lower performance compared with conventional horizontal bore systems.

**Table 3 pharmaceutics-14-00801-t003:** Advantages and limitations of magnetic marker monitoring.

Advantages	Limitations
Non-invasive.	Specialized equipment necessary.
Radiation-free.	No anatomical imaging.
High temporal and spatial resolution.	Low availability.
3D imaging.	Limited to the investigation of one dipole.

**Table 4 pharmaceutics-14-00801-t004:** Advantages and limitations of aspiration of luminal contents.

Advantages	Limitations
Only technique available to assess luminal drug concentrations and GI fluid composition.	Technique itself may alter GI physiology.
Mainly requires equipment that is commonly available at hospitals.	Composition of GI fluids may be altered (e.g., sampling procedure, storage conditions).
Combination with additional techniques (e.g., blood sampling, manometry, MRI) is possible.	Assessment of the drug amount available for absorption is not possible (owing to lack of information on fluid volumes).
Data generated provide important inputs for in vitro and in silico models, and can also be used to validate these models.	Difficult collection of fluids from the distal small intestine and colon impedes time-dependent assessment of luminal drug concentrations in the colon.
	Luminal concentrations are not necessarily the driving force for absorption due to interaction with intraluminal compounds.

**Table 5 pharmaceutics-14-00801-t005:** Advantages and limitations of intestinal perfusion tubes.

Advantages	Limitations
A suitable technique to determine intestinal site-specific permeability.	Some subjects may experience the GI tubes as uncomfortable.
Relevant in vivo parameter used to improve oral product development, and fundamental for the establishment of the BCS regulatory framework.	Difficult to reach and perfuse the distal small intestine and colon.
Possibility to determine permeability by both intestinal disappearance and drug appearance in the plasma compartment.	Some drugs can be significantly adsorbed by tube materials, and permeability cannot be assessed accurately.
Data generated have provided important inputs for validation of in vitro and in silico models commonly applied today.	GI tube itself may alter GI physiology.
Use of mainly gastroenterology equipment that is commonly available at university hospitals.	
Can be used to investigate intestinal and/or biliary excretion of drugs and formed metabolites.	

**Table 6 pharmaceutics-14-00801-t006:** Telemetric capsule systems.

Telemetric Capsule System	Functionality	Possible Application in Oral Biopharmaceutics
*SmartPill^TM^*	Measurement of pH, temperature and pressure.	Characterization of luminal pH values and motility (input for in vitro and in silico models).
*Heidelberg pH capsule*	Measurement of pH and temperature.	Characterization of luminal pH values (input for in vitro and in silico models).
*IntelliCap^®^*	Measurement of pH and temperature as well as dosing unit.	Investigation of regional absorption (e.g., as part of ER formulation development).
*PillCam^TM^*	Video recordings.	Visualization of formulation behavior and luminal environment.

**Table 7 pharmaceutics-14-00801-t007:** Advantages and limitations of telemetric capsules.

Advantages	Limitations
Move freely within the GI tract.	Drift of pH sensors is common.
Minimally invasive.	Single-use systems.
Measurements with high temporal resolution.	Lack of tracking.
	Large dimensions (swallowability problems, potential effects on transit times).

**Table 8 pharmaceutics-14-00801-t008:** Advantages and limitations of high-resolution manometry.

Advantages	Limitations
The only technique able to measure different motor patterns and their characteristics in each segment of the gut.	Positioning of a manometry catheter can be challenging, and requires training.
Evaluation of the effect of esophageal motility in transit of different formulations is possible through less invasive esophageal manometry.	Colonic manometry requires prior bowel preparation with oral laxatives or enema, and is normally performed during colonoscopy, which requires an endoscopy unit with trained personnel.
Water-perfused manometry catheters can be combined with GI sampling.	Interpretation of manometric recording requires some training.

**Table 9 pharmaceutics-14-00801-t009:** Overview of established pharmacokinetic marker substances and their applications.

Marker Substance	Possible Applications	Dosing Recommendations	Matrix
*Paracetamol (PCM)*	Determination of gastric emptying of liquids.	1.5 g of PCM as solution, tablet, or mixed with food.	Blood plasma or saliva.
*Caffeine* *(CAF)*	(a) Determination of gastric emptying of liquids; (b) Determination of in vivo disintegration time of oral dosage forms.	(a) 35 mg of CAF in an “ice capsule”; (b) 25–50 mg of CAF embedded into the dosage form.	Saliva.
*Sulfamethizole (SFM)*	Determination of gastric emptying of digestible solids.	15 capsules with a total amount of 3 g of SFM (in a matrix out of chicken egg albumin, Arabic gum powder, and distilled water).	Blood plasma.
*Sulfasalazine (SFS)*	Determination of orocecal transit time.	250–3000 mg of SFS in immediate-release dosage form or mixed with food.	Blood plasma or saliva.

**Table 10 pharmaceutics-14-00801-t010:** Advantages and limitations of pharmacokinetic marker substances.

Advantages	Limitations
Low costs and easy accessibility.	Indirect measurements.
No influence on GI physiology.	Prone to pharmacokinetic interactions.
Investigations under physiological conditions.	Abstinence of the marker substance is required.
When saliva may be used as specimen (i.e., paracetamol, caffeine, sulfasalazine): Non-invasive, high temporal resolution, small sample amounts (<1 mL) sufficient.	Side effects of the marker substance are possible.
Well accepted by subjects/patients.	

## Data Availability

Not applicable.

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
