# Peer review of "Application of In Vivo Imaging Techniques and Diagnostic Tools in Oral Drug Delivery Research"

_pharmaceutics, 2022, doi:10.3390/pharmaceutics14040801_

Round 1

Reviewer 1 Report

It is a very interesting article that addresses a topic of relevance in the pharmaceutical field to evaluate oral pharmaceutical forms in the body using imaging and diagnostic techniques. There are only minimum points to consider:

1) It would be highly recommended to include general information on the structure of the gastrointestinal tract before entering fully into the techniques used in oral biopharmaceutics.

2) Quality of figure 1 needs to be improved. Legends are not clearly readable.

3) In page 5, line 1 200. Chapter 0 is correct?

4) It would be very convenient to include a general table of the advantages and disadvantages offered by each technique and the drugs that have been evaluated with them.

Reviewer 2 Report

The authors have done extensive literature review on the topic and nicely put together the contents of this manuscript. However, a major lack is observed in the English grammar and sentence formation. This requires thorough check and editing.

The references have been divided at each technique which makes it lose uniformity and it would be advisable to add all the references at the end of the manuscript rather than at page breaks.

Reviewer 3 Report

The review paper is well written and of high interest to readers in the field. 

Only two minor aspects may be changed:

1) Reference to "chapters" (page 2, line 55) may be changed to "sections" since this is a review paper. Also, what is meant by "chapter 0" on page 5, line 200?

2) Page 2, line 58: Phase I clinical trials often don't measure pharmacodynamics since this phase make use of healthy volunteers.

Reviewer 4 Report

In this review, the most important in vivo techniques applied in oral biopharmaceutics are presented concisely in terms of potential applications as well as in terms of their advantages and limitations.

The paper is very well written. The examples are representative and references are slightly actual (minor comment: there is not one example of the year 2022).

As another minor comment: it would be useful to make a brief summary (a general comparison) about the potential of all technique for different pharmaceutical dosage forms (correlation in silico, in vitro and in vivo) and prediction capacity of each one.
